# Mamba-Enhanced Visual-Linguistic Representation for Multi-Label Image Recognition

**Zichang Tan**[1,2,*]    **Hao Tan**[3,*]    **Yang Yang**[3*]    **Prayag Tiwari**[4]    **Shifeng Chen**[1†]
**Jun Wan**[3]    **Xu Zhou**[2†]    **Zhen Lei**[3]

[1] *Shenzhen Institutes of Advanced Technology (SIAT), Chinese Academy of Sciences*    [2] *Sangfor Technologies Inc.*
[3] *Institute of Automation, Chinese Academy of Sciences*    [4] *Halmstad University*
*tanzichang@foxmail.com, tanhao2023@ia.ac.cn, shifeng.chen@siat.ac.cn, zhouxu@sangfor.com.cn*

**Reviewed on OpenReview:** *https://openreview.net/forum?id=KCz9Z9VNwr*

## Abstract

Multi-label image recognition stands as a foundational task in computer vision. Recently, vision-language models have made significant progress in this domain. However, previous approaches mostly utilized language models in a simplistic manner, without fully leveraging their potential. To address this, we propose a Mamba-enhanced Visual-Linguistic Representation (MVLR) framework for multi-label image recognition, which aims to better leverage the capabilities of the visual-linguistic representations. In our MVLR, we first propose a Prompt-Driven Label Representation learning (PDLR), which consists of both hard and soft prompts for acquiring comprehensive semantic knowledge for all labels from the large language model. After extracting the label representations, we propose an Interaction and Fusion Module (IFM) to interact with those representations and then fuse them together. To be specific, IFM first employs a label attention to explore the label co-occurrence relations and a context-aware attention to adaptively aggregate context information into label representations. Then, IFM further employs a channel attention to fuse the two features together, forming more reliable and effective label representations. Finally, we propose a Quadruplet Mamba-enhanced Visual- Linguistic block (QMVL) to mutually interact with visual and linguistic features with the strong structure of Mamba. Our QMVL simultaneously emphasizes the features of both visual and linguistic modalities, which is greatly different from previous works that take linguistic information as a secondary supplementary item. Extensive experiments on several popular datasets, including MS-COCO, Pascal VOC 2007 and NUS-WIDE, demonstrate the superiority of our MVLR.

## 1 Introduction

Multi-label recognition (MLR) (Chen et al., 2019b; Wang et al., 2020; Du et al., 2024) is a foundational yet challenging task in computer vision that enables comprehensive scene understanding through simultaneous identification of multiple labels within a single image. This capability holds transformative potential for critical applications including: (1) intelligent surveillance systems requiring real-time analysis of human attributes (Tan et al., 2020; Wu et al., 2022; Wang et al., 2022), (2) next-generation retrieval systems with semantic-aware search capabilities (Wei et al., 2024), and (3) medical image diagnosis where multi-label annotation improves diagnostic accuracy. Despite decades of research, current MLR systems still struggle with complex label correlations and semantic gaps, particularly in handling the interplay between visual and linguistic modalities - a fundamental limitation our work directly addresses.

With the emergence of vision-language pre-training techniques (Radford et al., 2021; Jia et al., 2021), many recent works (Chen et al., 2019b; You et al., 2020; Wang et al., 2020; Zhao et al., 2021; Zhu et al., 2022;

---

[*]Co-first author, [†]Corresponding author.

Li et al., 2023b) have utilized the linguistic modality to supplement semantic information into the visual features. Leveraging the rich semantic knowledge present in large language models, these approaches have shown enhancements in multi-label recognition tasks. Although current methods (Chen et al., 2019b; You et al., 2020; Wang et al., 2020; Zhao et al., 2021; Zhu et al., 2022; Li et al., 2023b) have made good progress for multi-label image recognition through utilizing visual-linguistic information, there are still some shortcomings. First, the knowledge extraction challenge persists in current paradigms. To be specific, the current methods all use rudimentary techniques to acquire knowledge of the large language models, failing to fully harness the potential of these expansive models. For example, most methods (Chen et al., 2019b; You et al., 2020; Wang et al., 2020; Zhu et al., 2022) rely on static label names as sole inputs to language models, generating rigid linguistic embeddings. This static approach fundamentally limits the model's capacity to capture dynamic semantic relationships. Second, the correlation modeling challenge emerges from isolated label processing. Existing frameworks extract label embeddings independently without modeling inter-label dependencies (e.g., "dog" → "animal" or "running" → "motion"), overlooking the hierarchical and compositional nature of semantic structures that could significantly boost recognition accuracy. Third, treating linguistic features as secondary supplements neglects their importance. While modern language models can produce discriminative embeddings (Radford et al., 2021), current architectures fail to establish equitable cross-modal interaction mechanisms, creating an information bottleneck that undervalues linguistic cues compared to visual features. Therefore, placing more emphasis on linguistic features and conducting more in-depth visual-linguistic interactions could largely enhance the model's capabilities.

In order to address the aforementioned problems, we propose a **M**amba-enhanced **V**isual-**L**inguistic **R**epresentation (MVLR) framework for multi-label image recognition. Inspired by previous works (Mehta et al., 2022; Yao et al., 2023), we formulate a Prompt-Driven Label Representations learning (PDLR) to tackle the first issue. The goal of PDLR is to extract reliable and comprehensive linguistic embeddings for all labels. To be specific, we take two kinds of prompts as inputs to the language model for acquiring semantic knowledge. One is hard prompts, where the prompts follow the hand-crafted templates like "`a photo of a` [CLS]`.`". "[CLS]" represents a label name like "`table`" or "`bird`". The other is soft prompts, where all prompts are set as learnable embeddings, which can be adaptively adjusted during the training process. Hard prompts are manually set and help to extract static but accurate linguistic embeddings for each label. Soft prompts are learnable embeddings guided by the training loss and help capture latent and necessary linguistic embeddings for all labels.

To address the second issue, an Interaction and Fusion Module (IFM) is further proposed to deeply aggregate the captured label representations. In the proposed IFM, we first apply label attention to explore the label co-occurrence among different labels based on the extracted label embeddings w.r.t. hard prompts. Then, we further employ a context-aware attention module based on the extracted label embeddings w.r.t. soft prompts to adaptively aggregate context information into label representations, which explores context-aware label attention. Specifically, we further propose a knowledge-to-context regularization (KCR) loss to constrain the two features from learning similar representations, further enhancing the generalization ability. Later, we further fuse the above two kinds of features using a channel interaction, which forms more effective, reliable and comprehensive label representations. As we can see, the core of our proposed IFM is to facilitate interactions among various label representations, establishing profound connections between different labels. As a result, the proposed IFM could overcome the limitation of the isolated process of extracting label features in PDLR.

To overcome the third problem, we further facilitate a mutual interaction between visual and linguistic modalities using the proposed Quadruplet Mamba-enhanced Visual-Linguistic block (QMVL), which is constructed on the Mamba structure (Zhu et al., 2024b) of good at modeling the sequence relations. Different from existing methods (Chen et al., 2019b; You et al., 2020; Wang et al., 2020; Zhu et al., 2022; 2023) where textual information is only unidirectionally integrated with visual information, our proposed QMVL allows a bidirectional interaction between the two modalities. In QMVL, we merge visual and linguistic features in both forward and inverse sequences to generate four concatenated features. Each feature pair corresponds to either the visual or linguistic modality, which is then fed into a Mamba block (Zhu et al., 2024b), for extracting the respective features. In this module, the Mamba structure is taken for visual-linguistic interaction due to its strong ability in relation modeling. Specifically, both forward and inverse sequences are employed

in the Mamba block, which aims to extract more order-independent relations and representations. Finally, unlike previous methods (Chen et al., 2019b; You et al., 2020; Wang et al., 2020; Zhu et al., 2022; 2023) that employ fixed classification weights (e.g., linear layers) to generate the final predictions, our method predicts the scores based on the similarity between these two representations (i.e., the dot product of visual and linguistic representations), where the visual features represent the input features and linguistic features denote the class center of each label. In this way, it achieves *input-adaptive category centers*, largely enhancing the model's generalization capability. To sum up, the main contributions of this work include:

- We propose the Mamba-enhanced Visual-Linguistic Representation (MVLR), a novel visual-linguistic representation learning framework for multi-label image recognition, which achieves state-of-the-art performance on multiple widely used benchmarks.

- We propose an Interaction and Fusion Module (IFM) to deeply aggregate the label representations. Specifically, multiple attention mechanisms including a label attention, a context-aware label attention and a channel attention are employed to capture label relations and establish profound connections between different labels.

- We propose a Quadruplet Mamba-enhanced Visual-Linguistic attention (QMVL) to conduct a bidirectional interaction between visual and linguistic modalities based on the Mamba structure. The proposed QMVL simultaneously emphasizes the features of both modalities thereby maximizing the utilization of language models in the process.

## 2 Related Work

### 2.1 Multi-Label Classification

Multi-label recognition (Wang et al., 2016; Liu et al., 2022) is a crucial task within computer vision. Early methodologies (Ye et al., 2020; Lanchantin et al., 2021) primarily focused on a single visual modality as the input, often employing Recurrent Neural Networks (RNN) and graph-based models for modeling the label relations. For instance, Wang et al. (Wang et al., 2016) investigate semantic correlations by integrating RNN with the feature extractor, while some other researchers (Ye et al., 2020; Tan et al., 2020) consider using the graph convolutional network (GCN) to capture the relations among different labels. Recently, transformer (Vaswani et al., 2017) has been proposed and demonstrated strong abilities in relation modeling especially for long sequences (Lanchantin et al., 2021; Weng et al., 2023). Recently, inspired by the success of vision-language models, PatchCT (Li et al., 2023b) leverages the rich semantic priors from CLIP to implicitly capture label relationships. Some works (Tan et al., 2025; Li et al., 2023b) also utilize the Optimal Transport to model the relationships between image features and labels.

Most previous methods (Tan et al., 2019; 2020) solely rely on the visual modality, resulting in a limited capacity of the model in semantic understanding. Recently, a large number of language models (e.g., (Devlin et al., 2019)) have been proposed and have achieved great success in related fields. Many researchers (Chen et al., 2019b;c; Wang et al., 2020; You et al., 2020; Liu et al., 2021; Zhu et al., 2022; Zhao et al., 2021; Zhu et al., 2023; Li et al., 2023b) also realize the importance of language modality and gradually use both visual and linguistic modalities to address multi-label image recognition problems, usually incorporating linguistic modalities to enrich semantic information. For example, (You et al., 2020) propose a cross-modality attention module to merge visual features and label embeddings.

It is a feasible approach to leverage both visual and linguistic modalities for addressing multi-label image recognition. For example, (Wang et al., 2024a) propose a CLIP-guided vision-language fusion framework for pedestrian attribute recognition, where several transformer blocks are employed to fuse vision and language features for final recognition. Our proposed method is developed based on the visual-linguistic features. Compared to previous works, we aim to achieve a more in-depth visual-linguistic representation learning by the proposed PDLR, IFM and QMVL, which targets extracting effective label representations, exploring in-depth label relations, and conducting comprehensive interactions, respectively.

## 2.2 Vision-Language Models

The utilization of large-scale vision-language pre-training has emerged as a potent strategy spanning a diverse array of visual tasks (Sun et al., 2022; Radford et al., 2021). Vision-language models (VLMs) (Radford et al., 2021; Jia et al., 2021) harness a contrastive-based pre-training methodology to forge a cohesive representation uniting visual and linguistic elements, building a strong connection between the two modalities. Many researchers have achieved good applications in downstream tasks using vision-language models (Sun et al., 2022). For example, CoOp (Zhou et al., 2022b) replaces manual prompts with learnable context vectors for efficient adaptation, while CoCoOp (Zhou et al., 2022a) extends this by generating instance-conditional prompts to improve generalization on unseen classes. DualCoOp (Sun et al., 2022) leverages the CLIP model (Radford et al., 2021) for multi-label image recognition for achieving fast adaption by only using limited annotations. Similarly, to address the problem of multi-label image recognition, (Ding et al., 2023) explore the structured semantic prior based on the extracted linguistic embeddings, which facilitates the task of multi-label image recognition. Recently, there has been a trend among researchers to merge pre-trained language and vision models to create robust Vision-Language Models (VLMs). Examples of such models like Flamingo (Alayrac et al., 2022), BLIP-2 (Li et al., 2023a), MiniGPT-4 (Zhu et al., 2024a) and LLaVA (Liu et al., 2024a). These models have demonstrated impressive capabilities across various visual tasks. For example, Flamingo (Alayrac et al., 2022) introduced gated attention mechanisms for modality interactions, showcasing promising few-shot learning capabilities. In this paper, our proposed method is also constructed based on vision-language models, achieving reliable multi-label image recognition by learning robust visual-linguistic representations.

## 2.3 State Space Model

Being at the forefront of the State-Space Models (SSMs) era, Gu et al. (Gu et al., 2022; Wang et al., 2024b) introduced the innovative Structured State-Space Sequence (S4) model, which serves as an alternative to convolutional neural networks (CNNs) or transformers for effectively capturing long-range dependencies. A recent advancement by the work (Smith et al., 2023) introduces the S5 layer, which incorporates MIMO SSM and efficient parallel scanning into the existing S4 layer architecture. Concurrently, (Fu et al., 2023) introduces the H3 SSM layer, further enhancing the efficacy of SSMs. In a recent development, (Gu & Dao, 2024) introduced a versatile language model known as Mamba. Specifically, Mamba surpasses transformers in performance metrics across diverse scales of extensive real-world data. The research of Mamba has piqued the research interest of many researchers, where lots of Mamba-based works are proposed and achieved promising applications in various tasks, like Vision Mamba (Zhu et al., 2024b; Wang et al., 2024b). More works about state-space models can be found in related surveys, like (Wang et al., 2024b), VideoMamba (Li et al., 2024), Pointmamba (Liang et al., 2024), Swin-umamba (Liu et al., 2024b), and so on. In this study, we aim to extend the application of Mamba to the domain of multi-label image recognition. The primary focus is on harnessing the robust structure of Mamba to facilitate visual-linguistic interactions.

## 3 Proposed Method

The overall pipeline of our MVLR framework is shown in Figure 1. In the following sections, we will introduce PDLR, IFM and QMVL in detail.

### 3.1 Preliminary

**Notations.** For multi-label image recognition, assume the input image $\boldsymbol{I} \in \mathbb{R}^{H \times W \times 3}$ is labeled with $C$ candidate categories, where $\boldsymbol{y} \in \mathbb{R}^C$ represents the multi-hot label vector and $\boldsymbol{y}_j = 1$ means the input image contains the $j^{th}$ label and vice versa. For the input image $\boldsymbol{I}$, we employ an image encoder (e.g., ResNet (He et al., 2016) or ViT (Wang et al., 2021)) to extract visual features $\boldsymbol{X} \in \mathbb{R}^{M \times d}$, where $M$ indicates the number of pixels or patches, and $d$ is the feature dimension.

**Attention mechanism.** Transformer has achieved significant success in visual tasks (Wang et al., 2021; Chen et al., 2021; Arnab et al., 2021), particularly due to its well-designed attention mechanism, which exhibits strong capability in relation modeling. Typically, there are two kinds of attention mechanisms in

the transformer, i.e., self-attention and cross-attention. For self-attention, it models the relations among the elements within an input sequence $\boldsymbol{E} \in \mathbb{R}^{M \times d}$ ($M$ is the number of vectors and $d$ denotes the dimension of the features), which is formulated as:

$$\text{Self-Attn}(\boldsymbol{E}) = \text{softmax}(\frac{\boldsymbol{Q}\boldsymbol{K}^{\mathsf{T}}}{\sqrt{d}})\boldsymbol{V},$$
$$\text{where } \boldsymbol{Q} = \boldsymbol{E}\boldsymbol{W}_Q, \boldsymbol{K} = \boldsymbol{E}\boldsymbol{W}_K, \boldsymbol{V} = \boldsymbol{E}\boldsymbol{W}_V, \tag{1}$$

where $\boldsymbol{W}_Q$, $\boldsymbol{W}_K$ and $\boldsymbol{W}_V$ are learnable weights. We take $\boldsymbol{\mathcal{M}} = \text{softmax}(\frac{\boldsymbol{Q}\boldsymbol{K}^{\mathsf{T}}}{\sqrt{d}})$ to denote the attention map that captures the pair-wise relations of vectors in $\boldsymbol{E}$. In contrast, cross-attention takes different sources as input and is good at capturing cross-domain interactions. Suppose the inputs are denoted as $\boldsymbol{E}$ and $\boldsymbol{Z}$, the process is formulated as:

$$\text{Cross-Attn}(\boldsymbol{E}, \boldsymbol{Z}) = \text{softmax}(\frac{\boldsymbol{Q}\boldsymbol{K}^{\mathsf{T}}}{\sqrt{d}})\boldsymbol{V},$$
$$\text{where } \boldsymbol{Q} = \boldsymbol{E}\boldsymbol{W}_Q, \boldsymbol{K} = \boldsymbol{Z}\boldsymbol{W}_K, \boldsymbol{V} = \boldsymbol{Z}\boldsymbol{W}_V. \tag{2}$$

**State Space Model.** State Space Models (SSM) (Gu et al., 2022) are designed around continuous systems that map a 1-D function or sequence, $x(t) \in R^L \rightarrow y(t) \in R^L$, through a hidden state $h(t) \in R^N$. Assume the evolution parameter of the system is denoted as $\mathbf{A}^{N \times N}$ and the projection parameters are denoted by $\mathbf{B}^{N \times 1}$ and $\mathbf{C}^{1 \times N}$, the SSM system model the input data via an ordinary differential equation (ODE) as following:

$$h'(t) = \mathbf{A}h(t) + \mathbf{B}x(t), \ y(t) = \mathbf{C}h(t). \tag{3}$$

In recent Mamba structures (Gu & Dao, 2024; Mehta et al., 2022), they approximate the continuous ODE through a discretization, where a timescale parameter $\Delta$ is employed to transform the continuous parameters $\mathbf{A}$, $\mathbf{B}$ to their discrete form $\overline{\mathbf{A}}$, $\overline{\mathbf{B}}$. To be specific, the typical transform method is zero-order hold (ZOH), which can be represented as follows:

$$\overline{\mathbf{A}} = exp(\Delta\mathbf{A}), \ \overline{\mathbf{B}} = (\Delta\mathbf{A})^{-1}(exp(\Delta\mathbf{A}) - \mathbf{I}) \cdot \Delta\mathbf{B}. \tag{4}$$

After obtaining the discrete parameters $\overline{\mathbf{A}}$ and $\overline{\mathbf{B}}$, the discretized version of ODE can be written as:

$$h_t = \overline{\mathbf{A}}h_{t-1} + \overline{\mathbf{B}}x_t, \ y_t = \mathbf{C}h_t. \tag{5}$$

## 3.2 Prompt-Driven Label Representations Learning

Large VLMs (Radford et al., 2021; Jia et al., 2021) typically encompass a wealth of semantic knowledge since the pre-training on large-scale image-text pairs. Therefore, by setting appropriate textual inputs for each label, the embeddings extracted by the language model will contain the underlying semantic relations between different labels. For example, the textual embeddings of "trees" and "lawn" are likely to be close, while "television" and "elephant" may be more distant. In the realm of large language models, these suitable textual inputs are referred to as "prompts". Inspired by previous works (Mehta et al., 2022; Yao et al., 2023), we formulate a Prompt-Driven Label Representations learning (PDLR) for obtaining robust linguistic embeddings for all labels in this section. In the proposed PDLR, two kinds of prompts, namely hard and soft prompts, are taken as inputs for the large language model.

**Hard prompts.** For the hard prompts, we take the hand-crafted templates (i.e., hard prompts) "*This photo contains [CLS].*" as the inputs for all $C$ labels. Then the text encoder is adopted to extract $C$ label embeddings w.r.t the hard prompts, which are denoted as $\boldsymbol{T}^{hard} = \{\boldsymbol{t}_1^{hard}, ..., \boldsymbol{t}_C^{hard}\} \in \mathbb{R}^{C \times d}$, where $d$ is the hidden dimension of CLIP.

**Soft prompts.** Based on the above hard prompts, the networks could extract accurate linguistic embeddings for all labels. However, all prompts are manually set and will not be changed according to the training loss. Inspired by (Zhou et al., 2022b;a), we also utilize a set of soft prompts to extract label embeddings, where all prompts are learnable embeddings and will be adjusted to the optimal representations during the training

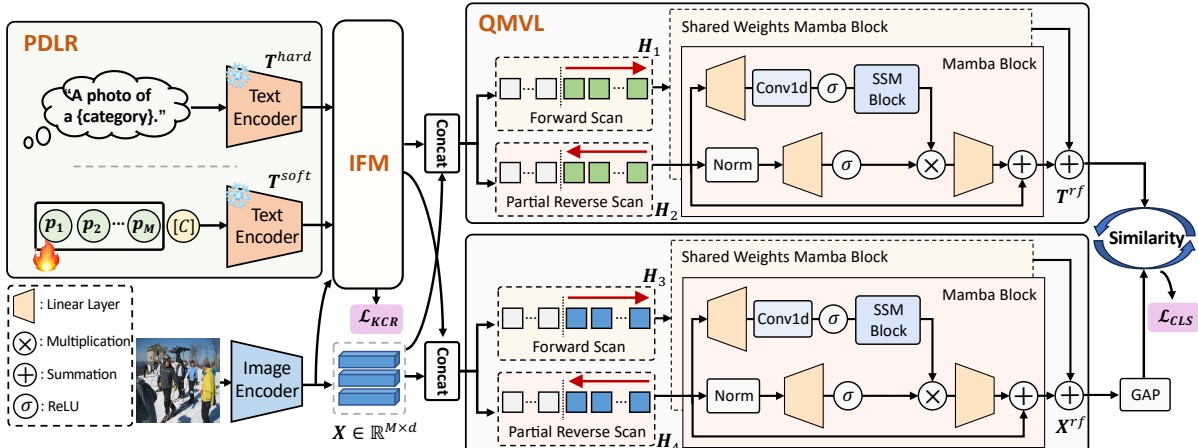

Figure 1: **Overview of the proposed MVLR framework. 1)** PDLR adopts both hard prompts and soft prompts to model label representations. **2)** IFM deeply aggregates the label embeddings from hard and soft prompts with several attentions and interactions. And $\mathcal{L}_{KCR}$ is measured to enhance the generalization. **3)** QMVL employs a quadruplet mamba structure to perform cross-modal interaction. The context-aware label representations are then regarded as the classification weights and the prediction is based on the similarity between these two representations, which achieves input-adaptive category centers.

process. To be specific, we prepend $L$ prompt tokens to each label and yield "$[\boldsymbol{p}_1][\boldsymbol{p}_2]...[\boldsymbol{p}_L][\boldsymbol{s}_j]$", where $\boldsymbol{p}_l \in \mathbb{R}^d$ is learnable to adapt the task and $\boldsymbol{s}_j$ is the word embedding of the $j^{th}$ label name. Then the sequences are fed into text encoder to extract $C$ label embeddings, which are denoted as $\boldsymbol{T}^{soft} = \{\boldsymbol{t}_1^{soft}, ..., \boldsymbol{t}_C^{soft}\} \in \mathbb{R}^{C \times d}$.

### 3.3 Interaction and Fusion Module

To fully leverage the strengths of the extracted label representations, we propose an Interaction and Fusion Module (IFM). As shown in Figure 2, in the proposed IFM, we first explore the label relations among the extracting label representations $\mathbf{T}^{ka}$ and $\mathbf{T}^{ca}$. To be specific, for the labels embeddings $\mathbf{T}^{ka}$ (w.r.t the hard prompts), we employ **label attention** to explore label co-occurrence among different labels. For the label embeddings $\mathbf{T}^{ca}$ w.r.t the soft prompts, we further propose a **context-aware attention** module to adaptively aggregate context information into label representations and model context-aware label relations. Later, we employ a **channel attention** to interact with the two kinds of label representations. Finally, we propose a **Relation Aggregation** to aggregate the relation-enhanced label representations together.

**Label attention.** We perform a label attention on the extracted label embeddings $\boldsymbol{T}^{hard}$ (w.r.t hard prompts) to capture the knowledge-guided relations, which is formally written as:

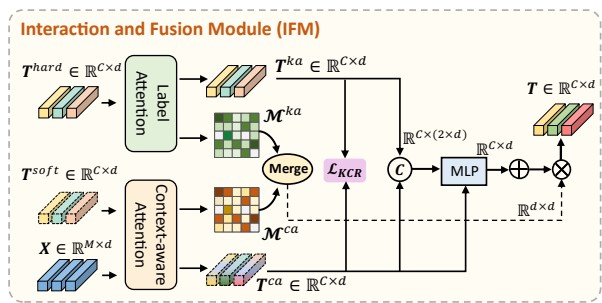

Figure 2: **Illustration of the Interaction and Fusion Module (IFM).** Label attention and Context-aware attention are first applied to capture label relations and cross-modal dependencies, respectively. Then $\mathbf{T}^{ka}$ and $\mathbf{T}^{ca}$ are deeply aggregated through channel interaction. Two relation maps are then gathered to refine the final label representations.

$$\boldsymbol{T}^{ka}, \boldsymbol{\mathcal{M}}^{ka} = \text{Self-Attn}(\boldsymbol{T}^{hard}), \tag{6}$$

where $\boldsymbol{T}^{ka} \in \mathbb{R}^{C \times d}$ is the relation-enhanced label embeddings. $\boldsymbol{\mathcal{M}}^{ka} \in \mathbb{R}^{C \times C}$ denotes the knowledge-guided attention map, where $\boldsymbol{\mathcal{M}}_{ij}^{ka}$ depicts the relation between $\boldsymbol{t}_i^{hard}$ and $\boldsymbol{t}_j^{hard}$, indicating the underlying co-occurrence probability of the $i^{th}$ and the $j^{th}$ label.

**Context-aware attention.** As mentioned above, the proposed label attention could extract rich semantic knowledge and relations. However, such information is static and independent of the input image. To address this, we further propose a context-aware attention module to capture context-aware label relations based on the image features and label embeddings w.r.t soft prompts. To be specific, we condition the label embeddings $\boldsymbol{T}^{hard}$ (w.r.t soft prompts) on visual features, which is formulated as:

$$
\begin{aligned}
\boldsymbol{T}^{'soft} &= \text{Cross-Attn}(\boldsymbol{T}^{soft}, \boldsymbol{X}), \\
\boldsymbol{T}^{ca}, \boldsymbol{\mathcal{M}}^{ca} &= \text{Self-Attn}(\boldsymbol{T}^{'soft}),
\end{aligned}
\tag{7}
$$

where $\boldsymbol{T}^{'soft}$ captures the interaction between each label representation and all spatial regions. $\boldsymbol{T}^{ca}$ is the context-aware label representations and $\boldsymbol{\mathcal{M}}^{ca}$ represents the context-aware attention map. In this way, fine-grained context clues would be incorporated into static label semantics, which makes the extracted label embeddings more reliable.

**Channel interaction.** To deeply integrate the relation-enhanced label representations $\boldsymbol{T}^{ka}$ and $\boldsymbol{T}^{ca}$, specifically, we propose a channel interaction between them to continually inject general knowledge into $\boldsymbol{T}^{ca}$, which is formulated as:

$$
\boldsymbol{T}^{ca} = \boldsymbol{T}^{ca} + \text{MLP}([\boldsymbol{T}^{ka}, \boldsymbol{T}^{ca}]),
\tag{8}
$$

where $\text{MLP}(\cdot)$ denotes a Multi-Layer Perceptron and $[\cdot]$ denotes the concatenation operation. For simplicity, we reuse the notation $\boldsymbol{T}^{ca}$ for the modulated embedding. Moreover, soft prompting is potential to overfit the seen data and forget the general knowledge (Bulat & Tzimiropoulos, 2023; Yao et al., 2023). Therefore, we further introduce a knowledge-to-context regularization (KCR) loss to enhance the generalization ability, which is formulated as:

$$
\mathcal{L}_{KCR} = \frac{1}{C} \sum_{j=1}^{C} \left(1 - \frac{\boldsymbol{t}_j^{ka}(\boldsymbol{t}_j^{ca})^{\mathsf{T}}}{||\boldsymbol{t}_j^{ka}||||\boldsymbol{t}_j^{ca}||}\right).
\tag{9}
$$

The goal of KCR is semantic alignment, rather than obtaining diversity features. The KCR loss prevents the two branches from drifting apart while allowing them to contain complementary information, which facilitates the subsequent feature fusion. Both $\boldsymbol{T}^{ca}$ and $\boldsymbol{T}^{ka}$ are label embeddings extracted through distinct methods. By minimizing the distance between them via Eq. 9, these embeddings converge to similar representations. This alignment reduces intra-class variance for each label, thereby enhancing model generalization.

**Relation aggregation.** Besides, the relation among labels should consider both general knowledge and practical contexts. Therefore, we propose to aggregate knowledge-guided attention map $\boldsymbol{\mathcal{M}}^{ka}$ and context-aware attention map $\boldsymbol{\mathcal{M}}^{ca}$ through a re-weighting scheme. The aggregated map is then adopted to enhance the label representations:

$$
\boldsymbol{T} = (\alpha\boldsymbol{\mathcal{M}}^{ka} + (1-\alpha)\boldsymbol{\mathcal{M}}^{ca})\boldsymbol{T}^{ca},
\tag{10}
$$

where $\alpha$ is set to be learnable and $\boldsymbol{T} \in \mathbb{R}^{C \times d}$ denotes the relation-enhanced label representations.

### 3.4 Quadruplet Mamba-Enhanced Visual-Linguistic Attention

While we have established prompt-driven label representations, the mutual interaction between visual and linguistic modalities remains underexplored. Inspired by recent advancements in state space models (Zhu et al., 2024b), we propose the Quadruplet Mamba-enhanced Visual-Linguistic (QMVL) attention module to facilitate deep cross-modal interaction. The standard Mamba architecture enables efficient $O(N)$ sequence modeling, but its causal scan is strictly unidirectional: in a concatenated sequence $[A, B]$, modality $B$ can condition on modality $A$, whereas $A$ cannot access $B$. Thus, directly applying Mamba to a single visual-linguistic sequence would yield order-dependent and one-sided cross-modal interaction. QMVL addresses this by decomposing the interaction into four complementary causal paths, constructing a virtual global receptive field within the linear-complexity Mamba framework.

Table 1: **Comparison (%) to state-of-the-art methods on MS-COCO.** Results with different backbone and input resolution are reported. Among them, mAP, OF1, and CF1 are the primary metrics (highlighted in **red**) as the others may be significantly affected by the threshold.

| Method | Backbone | Resolution | mAP | ALL | | | | | | Top-3 | | | | | |
|---|---|---|---|---|---|---|---|---|---|---|---|---|---|---|---|
| | | | | CP | CR | CF1 | OP | OR | OF1 | CP | CR | CF1 | OP | OR | OF1 |
| ML-GCN (Chen et al., 2019c) | ResNet101 | (448, 448) | 83.0 | 85.1 | 72.0 | 78.0 | 85.8 | 75.4 | 80.3 | 89.2 | 64.1 | 74.6 | 90.5 | 66.5 | 76.7 |
| CMA (You et al., 2020) | ResNet101 | (448, 448) | 83.4 | 82.1 | 73.1 | 77.3 | 83.7 | 76.3 | 79.9 | 87.2 | 64.6 | 74.2 | 89.1 | 66.7 | 76.3 |
| ASL (Ridnik et al., 2021) | ResNet101 | (448, 448) | 85.0 | - | - | 80.3 | - | - | 82.3 | - | - | - | - | - | - |
| Q2L-R101 (Liu et al., 2021) | ResNet101 | (448, 448) | 84.9 | 84.8 | 74.5 | 79.3 | 86.6 | 76.9 | 81.5 | 78.0 | 69.1 | 73.3 | 80.7 | 70.6 | 75.4 |
| SALGL (Zhu et al., 2023) | ResNet101 | (448, 448) | 85.8 | **87.2** | 74.5 | 80.4 | **87.8** | 77.6 | 82.4 | **90.4** | 65.7 | 76.1 | **91.9** | 67.9 | 78.1 |
| MambaML (Zhu et al., 2025) | ResNet101 | (448, 448) | 85.7 | 86.1 | 75.1 | 80.3 | 87.7 | 77.6 | 82.3 | 89.4 | 66.5 | 76.3 | 91.5 | 68.2 | 78.1 |
| **MVLR** | ResNet101 | (448, 448) | **88.5** | 83.1 | **82.5** | **82.8** | 83.5 | **85.3** | **84.4** | 88.7 | **69.4** | **77.9** | 90.5 | **71.3** | **79.8** |
| SSGRL (Chen et al., 2019b) | ResNet101 | (576, 576) | 83.6 | **89.5** | 68.3 | 76.9 | **91.2** | 70.7 | 79.3 | **91.9** | 62.1 | 73.0 | **93.6** | 64.2 | 76.0 |
| C-Tran (Lanchantin et al., 2021) | ResNet101 | (576, 576) | 85.1 | 86.3 | 74.3 | 79.9 | 87.7 | 76.5 | 81.7 | 90.1 | 65.7 | 76.0 | 92.1 | 71.4 | 77.6 |
| ADD-GCN (Ye et al., 2020) | ResNet101 | (576, 576) | 85.2 | 84.7 | 75.9 | 80.1 | 84.9 | 79.4 | 82.0 | 88.8 | 66.2 | 75.8 | 90.3 | 68.5 | 77.9 |
| Q2L-R101 (Liu et al., 2021) | ResNet101 | (576, 576) | 86.5 | 85.8 | 76.7 | 81.0 | 87.0 | 78.9 | 82.8 | 90.4 | 66.3 | 76.5 | 92.4 | 67.9 | 78.3 |
| SALGL (Zhu et al., 2023) | ResNet101 | (576, 576) | 87.3 | 87.8 | 76.8 | 81.9 | 88.1 | 79.5 | 83.6 | 91.1 | 66.9 | 77.2 | 92.4 | 69.0 | 79.0 |
| MambaML (Zhu et al., 2025) | ResNet101 | (576, 576) | 86.7 | 86.9 | 76.4 | 81.3 | 88.0 | 79.0 | 83.2 | 90.0 | 67.1 | 76.9 | 91.9 | 68.8 | 78.7 |
| **MVLR** | ResNet101 | (576, 576) | **89.0** | 83.0 | **83.7** | **83.3** | 83.7 | **86.8** | **85.2** | 89.3 | **70.2** | **78.6** | 91.5 | **72.0** | **80.6** |
| M3TR (Zhao et al., 2021) | ViT-B/16 | (448, 448) | 87.5 | **88.4** | 77.2 | 82.5 | **88.3** | 79.8 | 83.8 | **91.9** | 68.1 | 78.2 | **92.6** | 69.6 | 79.4 |
| PatchCT (Li et al., 2023b) | ViT-B/16 | (448, 448) | 88.3 | 83.3 | 82.3 | 82.6 | 84.2 | 83.7 | 83.8 | 90.7 | 69.7 | 78.8 | 90.3 | 70.8 | 79.8 |
| **MVLR** | ViT-B/16 | (448, 448) | **90.4** | 85.3 | **84.2** | **84.8** | 85.2 | **87.2** | **86.2** | 91.2 | **70.8** | **79.7** | 92.1 | **72.6** | **81.2** |

Given visual features $\mathbf{X}$ and label embeddings $\mathbf{T}$, we construct four sequences with different modality orders and intra-modality scan directions:

$$\begin{aligned}
\mathbf{H}_1 &= \texttt{concat}(\mathbf{X}, \mathbf{T}), \\
\mathbf{H}_2 &= \texttt{concat}(\mathbf{X}, \texttt{inv}(\mathbf{T})), \\
\mathbf{H}_3 &= \texttt{concat}(\mathbf{T}, \mathbf{X}), \\
\mathbf{H}_4 &= \texttt{concat}(\mathbf{T}, \texttt{inv}(\mathbf{X})),
\end{aligned} \tag{11}$$

where `concat` denotes concatenation and `inv` represents sequence reversal.

The quadruplet design has two roles. First, $\mathbf{H}_1$ and $\mathbf{H}_3$ swap the modality order, allowing label embeddings to be refined by preceding visual context and visual features by preceding linguistic context, thus approximating bidirectional cross-modal conditioning despite causal scanning. Second, $\mathbf{H}_2$ and $\mathbf{H}_4$ reverse only the modality being refined rather than the whole sequence, which preserves the conditioning prefix while capturing forward and backward intra-modality dependencies under the same cross-modal context.

The sequences $\mathbf{H}_1$ and $\mathbf{H}_2$ are processed by a shared Mamba block to refine the linguistic embeddings $\mathbf{T}^{rf}$, while $\mathbf{H}_3$ and $\mathbf{H}_4$ are processed by a separate Mamba block to refine the visual features $\mathbf{X}^{rf}$. The outputs from the normal and reversed pairs are summed to obtain order-robust visual-linguistic representations. The detailed architecture is illustrated in Fig. 1, with further technical details available in (Zhu et al., 2024b).

### 3.5 Visual-Linguistic Enhanced Training

Different from previous works (Chen et al., 2019b; You et al., 2020; Wang et al., 2020) that employ fixed classification weights (e.g., linear layers) for recognition, we regard each label linguistic representation as the center of the corresponding category, which is an input-adaptive approach and helps to enhance the model's generalization capability. To be specific, the presence probability of the $j^{th}$ label is predicted through measuring the similarity between visual representation $\boldsymbol{X}^{rf}$ and the $j^{th}$ label representation $\boldsymbol{T}_j^{rf}$:

$$\boldsymbol{p}_j = sigmoid(\boldsymbol{X}^{rf}(\boldsymbol{T}_j^{rf})^{\mathsf{T}}), \tag{12}$$

where $sigmoid(\cdot)$ is the sigmoid function to map the predicted logit into a probability. Based on final predictions in Eq. 12, the Asymmetric Loss (Ridnik et al., 2021) is employed for multi-label classification:

$$\mathcal{L}_{CLS} = \frac{1}{C} \sum_{j=1}^{C} \begin{cases} (1 - \boldsymbol{p}_j)^{\gamma^+} \log \boldsymbol{p}_j, & \boldsymbol{y}_j = 1, \\ \boldsymbol{p}_j^{\gamma^-} \log(1 - \boldsymbol{p}_j), & \boldsymbol{y}_j = 0, \end{cases} \tag{13}$$

Table 2: **Comparison (%) to state-of-the-art methods on Pascal VOC 2007.** Results are reported in terms of class-wise average precision (AP) and mean average precision (mAP). † indicates the ViT-B/16 backbone is used.

| Method | aero | bike | bird | boat | bottle | bus | car | cat | chair | cow | table | dog | horse | motor | person | plant | sheep | sofa | train | tv | mAP |
|---|---|---|---|---|---|---|---|---|---|---|---|---|---|---|---|---|---|---|---|---|---|
| SSGRL (Chen et al., 2019b) | 99.5 | 97.1 | 97.6 | 97.8 | 82.6 | 94.8 | 96.7 | 98.1 | 78.0 | 97.0 | 85.6 | 97.8 | 98.3 | 96.4 | 98.8 | 84.9 | 96.5 | 79.8 | 98.4 | 92.8 | 93.4 |
| ML-GCN (Chen et al., 2019c) | 99.5 | 98.5 | **98.6** | 98.1 | 80.8 | 94.6 | 97.2 | 98.2 | 82.3 | 95.7 | 86.4 | 98.2 | 98.4 | 96.7 | 99.0 | 84.7 | 96.7 | 84.3 | 98.9 | 93.7 | 94.0 |
| ASL (Ridnik et al., 2021) | - | - | - | - | - | - | - | - | - | - | - | - | - | - | - | - | - | - | - | - | 94.4 |
| SALGL (Zhu et al., 2023) | **99.9** | **98.8** | 98.3 | 98.2 | 81.6 | 96.5 | 98.1 | 97.8 | 85.2 | 97.0 | 89.6 | 98.5 | **98.7** | 97.1 | 99.2 | 86.9 | 96.4 | **89.9** | **99.5** | **95.2** | 95.1 |
| MambaML (Zhu et al., 2025) | 99.8 | 98.6 | 97.8 | 98.0 | 82.8 | 96.3 | 98.1 | 98.3 | 84.0 | 96.7 | 88.3 | 98.2 | 98.6 | 96.8 | 99.0 | 87.5 | 96.8 | 89.8 | 99.2 | 95.1 | 95.0 |
| **MVLR** | 99.7 | 98.1 | 98.5 | **99.3** | **87.0** | **98.2** | **98.3** | **98.9** | **86.7** | **98.3** | 89.5 | **99.2** | **98.7** | **97.7** | **99.3** | **88.3** | **97.6** | 87.0 | 99.3 | 94.1 | **95.7** |
| Q2L-TRL (Liu et al., 2021) | 99.9 | 98.9 | 99.0 | 98.4 | 87.7 | 98.6 | 98.8 | 99.1 | 84.5 | 98.3 | 89.2 | 99.2 | 99.2 | **99.2** | 99.3 | 90.2 | 98.8 | 88.3 | 99.5 | 95.5 | 96.1 |
| M3TR† (Zhao et al., 2021) | 99.9 | 99.3 | 99.1 | 99.1 | 84.0 | 97.6 | 98.0 | 99.0 | 85.9 | 99.4 | 93.9 | 99.5 | 99.4 | 98.5 | 99.2 | 90.3 | 99.7 | 91.6 | 99.8 | 96.0 | 96.5 |
| PatchCT† (Li et al., 2023b) | **100.0** | **99.4** | 98.8 | **99.3** | 87.2 | 98.6 | 98.8 | 99.2 | 87.2 | 99.0 | **95.5** | 99.4 | **99.7** | 98.9 | 99.1 | **91.8** | 99.5 | **94.5** | 99.5 | 96.3 | 97.1 |
| **MVLR†** | **100.0** | 98.9 | **99.4** | 99.1 | **91.2** | **99.5** | **98.9** | **99.5** | **91.1** | **99.7** | 93.1 | **99.8** | 99.6 | 98.7 | **99.4** | 90.5 | **99.9** | 91.0 | **99.9** | **96.7** | **97.3** |

where $\gamma^+$ and $\gamma^-$ are asymmetric focusing parameters for positive and negative samples, respectively.

Together with the hard-to-soft regularization loss, the final objective is defined as:

$$\mathcal{L} = \mathcal{L}_{CLS} + \lambda \mathcal{L}_{KCR}, \tag{14}$$

where $\lambda$ is a hyper-parameter to make a trade-off between the two losses.

# 4 Experiments

## 4.1 Datasets and Metrics

We employ six datasets for evaluation in total. To be specific, MS-COCO (Lin et al., 2014), PASCAL VOC 2007 (Everingham et al., 2010) and NUS-WIDE (Chua et al., 2009) are adopted for general multi-label image recognition. The details of the datasets would be introduced in the Appendix. Following the previous works (Wang et al., 2016; Chen et al., 2019c; Zhu et al., 2023), the mean average precision (mAP) is reported to evaluate the overall performance. Besides, we also report Class-wise Precision (CP), Recall (CR), F1 (CF1), and the average Overall Precision (OP), Recall (OR), F1 (OF1). Note that "CF1" and "OF1" are more informative since Precision and Recall vary with the threshold.

## 4.2 Implementation Details

Our method requires aligned visual and linguistic features simultaneously, which differs from some previous methods that rely solely on visual features. Therefore, we utilize CLIP (Radford et al., 2021) for extracting aligned textual embeddings and visual features. Our method requires aligned visual and linguistic features simultaneously. Therefore, we use CLIP (Radford et al., 2021) as the default visual and text encoder, since its image-text pre-training provides an aligned embedding space. We also evaluate other pre-trained vision-language models under the same ViT-B/16 backbone, with detailed results reported in Appendix Table 15. By default, we employ

Table 3: **Comparison (%) to state-of-the-art methods on NUS-WIDE.** † indicates ViT-B/16 backbone is used.

| Method | mAP | ALL | | Top-3 | |
|---|---|---|---|---|---|
| | | CF1 | OF1 | CF1 | OF1 |
| CMA (You et al., 2020) | 61.4 | 60.5 | 73.7 | 55.5 | 70.0 |
| ASL (Ridnik et al., 2021) | 63.9 | 62.7 | 74.6 | - | - |
| SALGL (Zhu et al., 2023) | 66.3 | 64.1 | 75.4 | 59.5 | 71.0 |
| MambaML (Zhu et al., 2025) | 65.9 | 63.7 | 75.0 | 59.8 | 70.7 |
| **MVLR** | **67.3** | **64.9** | **75.5** | **60.0** | **71.5** |
| Q2L-TRL (Liu et al., 2021) | 66.3 | 64.0 | 75.0 | - | - |
| PatchCT† (Li et al., 2023b) | 68.1 | 65.5 | 74.7 | 61.2 | 71.0 |
| **MVLR†** | **68.9** | **66.1** | **76.1** | **61.7** | **71.7** |

ResNet-101 (He et al., 2016) as the image encoder. The text encoder remains fixed during the training phase. The number of learnable prompt tokens, denoted as $L$, is configured to 4. Values for $\gamma^+$ and $\gamma^-$ are designated as 0 and 2, respectively. The hyper-parameter $\lambda$ is established as 4.0. Input images are resized to $448 \times 448$ during both the training and testing phases. The model is trained over 30 epochs using the AdamW optimizer with a batch size of 32. The learning rate is defined as 0.0001 and diminishes following a cosine policy. Consistent with (Ridnik et al., 2021; Zhu et al., 2023), we implement exponential moving average on model parameters with a decay rate of 0.9997.

## 4.3 Comparison with State-of-the-art

The comparisons on MS-COCO, PASCAL VOC 2007, and NUS-WIDE are shown in Table 1, Table 2 and Table 3, respectively. MVLR achieves state-of-the-art performance across various backbones and resolutions on all datasets, surpassing other methods with a decent margin. On the MS-COCO, compared with SALGL (Zhu et al., 2023) that utilized linguistic modality while hindering its role, MVLR exhibits considerable performance gains, exceeding them by 2.7% mAP. On the NUS-WIDE, our method surpasses all other methods on ResNet101 and ViT-B/16 backbones, achieving 67.3% and 68.9% mAP, respectively. Moreover, we also take PatchCT (Li et al., 2023b) for a fair comparison, which is also built upon the CLIP architecture with the same backbone. Our proposed MVLR also outperforms PatchCT on all three datasets, especially on MS-COCO dataset with 2.1% mAP improvements. The experimental results clearly confirm the effectiveness of our proposed method.

## 4.4 Ablation Studies

**Effect of proposed modules.** To verify the efefctiveness of the proposed PDLR, IFM and QMVL, we set a baseline method that utilizes pure category names to extract label representations and no further interactions are performed between modalities. As shown in Table 4, the performance of using pure label names is unsatisfactory, leading to poor CF1 and OF1. While PDLR improves the performance significantly. We attribute this to the fact that PDLR effectively extracts semantic knowledge from the text encoder. When combined with IFM, the performance achieves a large boost (+4.6% and +5.2% mAP on COCO and NUS respectively), sug-

Table 4: **Ablation study (%) on the proposed modules.** The baseline method in the first row utilizes the pure category names to extract label representations.

| PDLR | IFM | QMVL | MS-COCO | | | NUS-WIDE | | |
|---|---|---|---|---|---|---|---|---|
| | | | mAP | CF1 | OF1 | mAP | CF1 | OF1 |
| | | | 81.8 | 67.3 | 66.4 | 59.2 | 42.3 | 55.6 |
| ✓ | | | 83.4 | 76.8 | 80.7 | 60.6 | 55.0 | 73.5 |
| | | ✓ | 85.9 | 81.0 | 83.3 | 65.3 | 63.4 | 74.9 |
| ✓ | ✓ | | 88.0 | 82.4 | 84.1 | 65.8 | 63.7 | 74.9 |
| ✓ | | ✓ | 86.2 | 81.1 | 83.5 | 65.6 | 63.5 | 74.9 |
| ✓ | ✓ | ✓ | **88.5** | **82.8** | **84.4** | **67.3** | **64.8** | **75.3** |

gesting the effectiveness of deep fusion. Moreover, the proposed QMVL enhances the performance significantly, e.g., improve mAP by 4.1% and 6.1% on COCO and NUS compared to baseline. Notably, the three proposed modules work mutually and bring profound improvements when combined, which clearly demonstrates the effectiveness of all proposed modules.

**Scanning order in QMVL.** The mamba structure excels at processing 1-D sequences, and we aim to reduce the order-dependence of QMVL through appropriate scanning. As shown in Figure 3, there are typically four ways to scan the input sequences from two modalities. Figure 3 a) and Figure 3 b) view the concatenated sequence as a whole entity and b) further performs both forward and backward scan. Figure 3 c) and Figure 3 d) consider the difference of sequences and apply partial reverse scanning on

Table 5: **Ablation study (%) on the scanning order in QMVL.** Different orders are illustrated in Figure 3.

| Method | Order | MS-COCO | | | NUS-WIDE | | |
|---|---|---|---|---|---|---|---|
| | | mAP | CF1 | OF1 | mAP | CF1 | OF1 |
| Mamba | F. | 87.4 | 81.3 | 83.9 | 66.1 | 63.7 | 74.7 |
| Mamba | O-F.B. | 87.8 | 81.9 | 84.0 | 66.2 | 64.0 | 74.8 |
| Mamba | P-F.B.-S1 | 88.4 | 82.6 | 84.2 | 67.1 | 64.8 | 75.3 |
| Mamba | P-F.B.-S2 | **88.5** | **82.8** | **84.4** | **67.3** | **64.9** | **75.5** |

the former sequence and the latter sequence, respectively. From Table 5, partial reverse scanning (P-F.B.) performs significantly better than overall processing (O-F.B.). This is due to the fact that reversing the sequence independently is crucial for cross-modal interaction in mamba. Reversing the latter sequence (P-F.B.-S2) exhibits slightly better performance than the former (P-F.B.-S1), which indicates that bidirectional scanning of the current modality captures more effective interaction.

**Comparisons between attention-based structures and QMVL.** Attention-based structures are adept at processing inputs from different modalities. In this part, we compare the attention-based methods and our proposed QMVL. Specifically, we employ the standard self-attention block (Vaswani

Table 6: **Comparison (%) on the cross-modal interaction structure,** including attention-based and mamba structures.

| Method | FLOPs (G) | MS-COCO | | | NUS-WIDE | | |
|---|---|---|---|---|---|---|---|
| | | mAP | CF1 | OF1 | mAP | CF1 | OF1 |
| Cross-Attn. | 1.46 | 88.0 | 82.2 | 84.0 | 66.4 | 64.4 | 75.1 |
| Self-Attn. | 1.75 | 88.3 | 82.6 | 84.4 | 66.9 | 64.7 | 75.2 |
| QMVL | **1.17** | **88.5** | **82.8** | **84.4** | **67.3** | **64.9** | **75.5** |

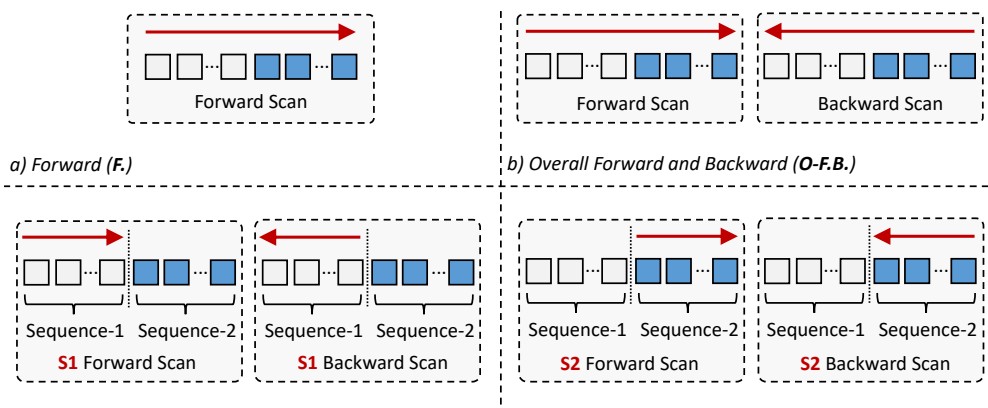

Figure 3: **Illustration of the scanning order in the cross-modal mamba blocks.** Typically, there are four ways to perform scanning for the two sequences. a) "F." and b) "O-F.B." treat the concatenated sequence as an overall entity while b) performs both forward and backward scan. c) "P-F.B.-S1" and d) "P-F.B.-S2" consider the difference of input sequences and perform partial reverse scanning. S1 and S2 come from different modalities.

et al., 2017) and cross-attention block (Vaswani et al., 2017) respectively. For the self-attention, inputs from different modalities are concatenated first. As shown in Table 6, self-attention performs better than cross-attention while suffering from larger FLOPs. Our proposed QMVL achieves better performance with lower FLOPs, which verifies that QMVL is more efficient at cross-modal interactions. We attribute this to the fact that our quadruplet design enables order-independent interactions while maintaining the efficiency of the mamba structure.

**Is using label representations as category centers better?** We verify the effectiveness of using label representations as category centers. As shown in Table 7, we set "Classifier Learning" as a reference, where we perform an uni-directional interaction and $C$ traditional classifiers are learned. Accordingly, "Label Rep." denotes the category centers are mapped from

Table 7: **Ablation study (%) on the generation of category centers.** "Label Rep." means the category centers are generated from label representations.

| Method | Text | MS-COCO | | | NUS-WIDE | | |
|---|---|---|---|---|---|---|---|
| | | mAP | CF1 | OF1 | mAP | CF1 | OF1 |
| Classifier Learning | CLIP-Text | 85.0 | 78.7 | 82.4 | 65.4 | 63.3 | 73.3 |
| Label Rep. | CLIP-Text | 81.6 | 54.6 | 47.7 | 58.6 | 29.1 | 36.6 |
| Label Rep. + QMVL | CLIP-Text | 85.9 | 81.0 | 83.3 | 65.3 | 63.4 | 74.9 |
| MVLR | CLIP-Text | **88.5** | **82.8** | **84.4** | **67.3** | **64.8** | **75.3** |
| | | (↑**3.5**) | (↑**4.1**) | (↑**2.0**) | (↑**1.9**) | (↑**1.5**) | (↑**2.0**) |
| Classifier Learning | BERT$_{Base}$ | 84.8 | 78.3 | 82.1 | 64.8 | 63.1 | 73.7 |
| Label Rep. | BERT$_{Base}$ | 83.6 | 78.1 | 81.8 | 60.2 | 57.5 | 73.6 |
| Label Rep. + QMVL | BERT$_{Base}$ | 85.7 | 80.8 | 83.0 | 64.9 | 63.1 | 74.6 |
| MVLR | BERT$_{Base}$ | **87.8** | **82.1** | **83.9** | **66.5** | **64.2** | **74.9** |
| | | (↑**3.0**) | (↑**3.8**) | (↑**1.8**) | (↑**1.7**) | (↑**1.1**) | (↑**1.2**) |

label representations, and "Label Rep. + QMVL" indicates that the QMVL is further employed. The low performance of "Label Rep." indicates that generating category centers directly from *static* label representations is inferior. However, this can be improved by constructing *dynamic* category centers through "Label Rep.+ QMVL". This indicates that QMVL can effectively model the cross-modal alignment and enable input-adaptive category centers. Moreover, MVLR surpasses all three reference methods, which further demonstrates the superiority of our proposed components including PDLR, IFM and QMVL. Our approach is robust to the choice of text encoder, and similar conclusions can be drawn when using BERT (Devlin et al., 2019) as the text encoder (shown in Table 7).

**Components of PDLR.** As shown in Table 8, employing solely hard prompts or soft prompts results in degraded performance, e.g., using only hard prompts leads to 1.0% mAP drop on both COCO and NUS-WIDE. Using more soft prompt tokens (e.g., $L = 8$) yields better results on MS-COCO but inferior results on the noisy NUS-WIDE. Therefore, we suggest using 4 prompt tokens in PDLR as a good trade-off.

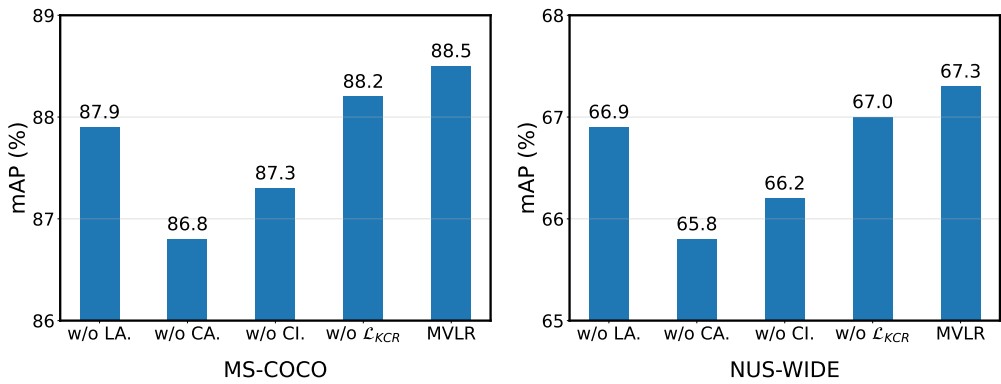

Figure 4: **Ablation study on the components of IFM.** "LA." denotes label attention. "CA." denotes context-aware attention. "CI." is channel interaction.

**Components of IFM.** As presented in Figure 4, context-aware attention plays a vital role in the proposed IFM, bringing 1.7% and 1.5% mAP improvements on COCO and NUS-WIDE respectively. This is due to the fact that context-aware attention significantly facilitates the integration of downstream semantics. The label attention, channel interaction and regularization loss $\mathcal{L}_{KCR}$ all help improve the performance and work mutually in our proposed MVLR, which clearly verifies the importance of each component in IFM.

## 5 Conclusion

In this work, we propose MVLR, a novel Mamba-enhanced visual-linguistic representation learning framework for multi-label image recognition. To address the defects of existing multi-modal approaches, we propose three modules, namely PDLR, IFM and QMVL to fully exploit the linguistic modality and learn the context-aware label representations and semantic-related visual representations concurrently. To be specific, the PDLR has utilized both hard and soft prompts for acquiring semantic knowl-

Table 8: **Ablation study (%) on the components of PDLR.** $L$ is the number of learnable tokens.

| Prompt Method | $L$ | MS-COCO | | | NUS-WIDE | | |
|---|---|---|---|---|---|---|---|
| | | mAP | CF1 | OF1 | mAP | CF1 | OF1 |
| Hard | - | 87.5 | 81.5 | 83.3 | 66.3 | 64.2 | 74.8 |
| Soft | 4 | 88.2 | 82.3 | 84.3 | 66.6 | 64.6 | 75.0 |
| Hard+Soft | 4 | 88.5 | **82.8** | 84.4 | **67.3** | **64.9** | **75.5** |
| Hard+Soft | 8 | **88.6** | **82.8** | **84.7** | 67.0 | 64.6 | 75.0 |
| Hard+Soft | 12 | 88.3 | 82.7 | 84.6 | 66.9 | 64.3 | 74.9 |

edge from the large language model, which helps the model to fully exploit the potential of linguistic modality. Then, the extracted label representations in PDLR have been aggregated by the IFM with several attention and interaction modules, aiming to capture more reliable and comprehensive label representations. Later, QMVL has conducted Mamba-enhanced visual-linguistic interactions among the visual and linguistic representations through a quadruplet of Mamba groups. We finally obtain the deeply-interacted, reliable and comprehensive visual and linguistic representations for multi-label image recognition. Extensive experiments show that Mamba-enhanced visual-linguistic representation learning is a reliable and useful way for multi-label image recognition, which achieves state-of-the-art performance on multiple widely used benchmarks.

**Broader Impact Statement** One limitation is that the semantic knowledge extracted by the pre-trained vision-language model relies on the model's pretraining data. This may introduce some unexpected noises to our method, and future work can explore the impact of these noises or other model biases on multi-label image recognition. Additionally, in our training datasets, there are unannotated objects present in the images, which could impact the model's performance in real-world scenarios. Our aim in this paper is to develop a general method for multi-label image recognition without targeting specific applications, which does not directly involve specific societal issues.

**Acknowledgments** This work was supported by the China Postdoctoral Science Foundation 2025M771717.

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

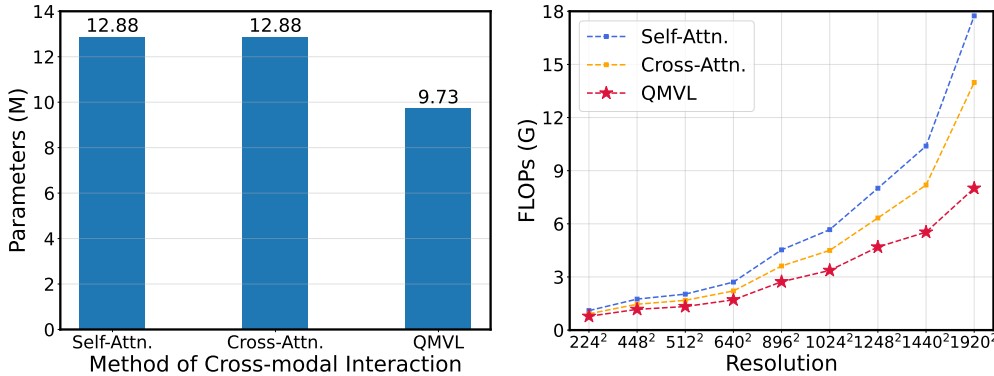

Figure 5: **Analysis on the number of parameters (left) and FLOPs (right)** of different methods of cross-modal interaction. Our QMVL has a reduced number of parameters compared to attention-based methods. QMVL exhibits a notable reduction in FLOPs when the input resolution increases.

Table 9: **Analysis on overall computational costs.** We report FLOPs, Parameters, Latency, and Peak Memory usage.

| Method | FLOPs (G) | Params. (M) | Latency (ms) | Peak Mem. (MB) | COCO | NUS |
|---|---|---|---|---|---|---|
| ML-Decoder (Ridnik et al., 2023) | 37.2 | 47.3 | 14.8 | 357.9 | 86.6 | 64.2 |
| Q2L (Liu et al., 2021) | 43.2 | 143.1 | 18.0 | 2369.3 | 84.9 | 65.0 |
| MambaVision (Hatamizadeh & Kautz, 2025) | 14.9 | 97.6 | 9.9 | 391.1 | 86.9 | 65.8 |
| SiMBA (Patro & Agneeswaran, 2024) | 39.1 | 62.5 | 21.4 | 336.3 | 82.3 | 61.9 |
| Baseline | 36.8 | 43.6 | 9.1 | 677.0 | 81.6 | 58.6 |
| **MVLR** | 37.6 | 47.8 | 13.1 | 677.2 | **88.5** | **67.3** |

# A  Appendix

## A.1  Details of Datasets

**MS-COCO.** The Microsoft COCO dataset (Lin et al., 2014) is widely utilized for evaluating multi-label classification tasks. It comprises a total of 123,287 images across 80 categories, with an average of approximately 2.9 labeled objects per image. We train the model on a training set containing 82,783 images and evaluate its performance on a test set consisting of 40,504 samples.

**PASCAL VOC 2007.** The VOC 2007 dataset (Everingham et al., 2010) serves as a widely used benchmark for multi-label recognition tasks, consisting of 9,963 images across 20 distinct object classes. On average, each image is annotated with 1.4 labels. Following the standard protocal (Chen et al., 2019c), we train the model on the train set containing 5,011 images and assess its performance on the test set comprising 4,952 images.

**NUS-WIDE.** NUS-WIDE (Chua et al., 2009) consists of 269,648 images annotated with 81 visual concepts and 5,018 labels. After removing unannotated samples, the dataset is randomly split into a training set and a test set, where the training set comprises 125,449 images and the test set contains 83,898 images. Noteworthy for its higher noise levels and increased difficulty compared to other benchmarks, NUS-WIDE presents unique challenges for evaluation.

## A.2  Further Analyses

**Analysis of computational costs.** Cross-modal interaction often suffers from heavy computational costs due to the need for fusing two large sequences, which is partially mitigated by our mamba-based structure. As presented in Figure 5, we compare the proposed QMVL and two attention-based methods (i.e., self-

Table 10: The comparisons on the MS-COCO dataset.

| Method | mAP | ALL | | | | | | Top-3 | | | | | |
|---|---|---|---|---|---|---|---|---|---|---|---|---|---|
| | | CP | CR | CF1 | OP | OR | OF1 | CP | CR | CF1 | OP | OR | OF1 |
| MambaVision-B | 86.9 | 86.2 | 74.7 | 80.0 | 87.4 | 76.3 | 81.5 | 90.3 | 66.9 | 76.8 | 91.6 | 68.2 | 78.2 |
| SiMBA-Base | 82.3 | 82.5 | 73.6 | 77.8 | 83.3 | 75.5 | 79.2 | 88.3 | 64.7 | 74.7 | 89.2 | 66.4 | 76.1 |
| **MVLR** | **88.5** | 83.1 | **82.5** | **82.8** | 83.5 | **85.3** | **84.4** | 88.7 | **69.4** | **77.9** | 90.5 | **71.3** | **79.8** |

Table 11: The comparisons on the NUS-WIDE dataset.

| Method | mAP | ALL | | Top-3 | |
|---|---|---|---|---|---|
| | | CF1 | OF1 | CF1 | OF1 |
| MambaVision-B | 65.8 | 64.6 | 74.7 | 59.7 | 70.4 |
| SiMBA-Base | 61.9 | 60.3 | 72.2 | 56.0 | 68.8 |
| **MVLR** | **67.3** | **64.9** | **75.5** | **60.0** | **71.5** |

attention block and cross-attention block). QMVL has a reduced number of parameters compared to both self-attention and cross-attention. Moreover, QMVL always exhibits lower FLOPs than attention-based structure, while the reduction in FLOPs is more significant when the input resolution increases. Note that QMVL achieves better performance (in Table 6) with lower computational costs, which indicates that QMVL is more efficient at cross-modal interaction. We also report the overall computational costs of our method. As shown in Table 9, compared to previous methods (Ridnik et al., 2023; Liu et al., 2021; Hatamizadeh & Kautz, 2025; Patro & Agneeswaran, 2024), our method achieves superior performance with highly efficient FLOPs and inference latency, and the inference latency is significantly lower than that of Q2L (Liu et al., 2021) and SiMBA (Patro & Agneeswaran, 2024).

**Comparisons to modern SSM methods.** In this section, we compare the proposed MVLR with two modern SSM methods, including MambaVision (Hatamizadeh & Kautz, 2025) and SiMBA (Patro & Agneeswaran, 2024). We re-implemented MambaVision-B and SiMBA-Base on MS-COCO and NUS-WIDE. As presented in Tables 10 and 11, these SSM baselines consistently underperform compared to our MVLR. Specifically, MVLR outperforms SiMBA by 6.2% mAP and 5.2% OF1 on the MS-COCO dataset, and by 5.4% mAP and 3.3% OF1 on the NUS-WIDE dataset. Furthermore, our method maintains a clear advantage over MambaVision, demonstrating the superior effectiveness of MVLR over existing SSM baselines.

**Sensitivity analysis of $\lambda$.** As shown in Figure 6, we evaluate the parameter sensitivity of $\lambda$ in Eq. 14. The results suggest that the performance of MVLR is generally stable. However, $\lambda > 4$ may result in a performance decline since an overly large $\lambda$ forces the learnable feature to collapse into the original feature space. Therefore, $\lambda = 4$ is a good trade-off.

Table 12: **Sensitivity analysis (%) of $\alpha$.** Results are reported on MS-COCO.

| $\alpha$ | 0.0 | 0.2 | 0.4 | 0.5 | 0.6 | 0.8 | 1.0 |
|---|---|---|---|---|---|---|---|
| mAP | 87.9 | 88.0 | 88.4 | 88.5 | 88.2 | 88.0 | 87.5 |
| CF1 | 82.3 | 82.2 | 82.6 | 82.8 | 82.7 | 82.4 | 82.0 |
| OF1 | 83.8 | 83.8 | 84.1 | 84.4 | 84.2 | 84.1 | 83.5 |

**Sensitivity analysis of $\alpha$.** As shown in Table 12, we analyze the influence of different choices of $\alpha$ in Eq. 10. A value between 0.4 and 0.6 achieves better performance than $\alpha = 0$, which verifies the effectiveness of relation aggregation. Increasing $\alpha$ does not scale up the performance, since the knowledge-aware relationships may shrink the learning abilities of soft prompts.

**Experiments on domain-specific scenarios.** To further verify the efficacy and robustness of our proposed method in handling domain-specific multi-label classification, we extend our experiments to remote sensing and medical imaging domains. **Remote Sensing.** We evaluate our method on two large-scale remote sensing benchmarks: MultiScene (Hua et al., 2021) and MLRSNet (Qi et al., 2020). As shown in Table 13, MVLR consistently outperforms the baseline RemoteCLIP-FT across different architectures (ResNet50 and ViT-B/32). Notably, our method achieves superior performance not only in mAP but also in both ALL and Top-3 F1 scores (CF1 and OF1), demonstrating its strong capability in recognizing complex aerial scenes. **Medical Imaging.** Furthermore, we assess our method in the medical imaging domain using the ChestX-ray14 dataset (Wang et al., 2017). To ensure a fair comparison with existing approaches, we strictly

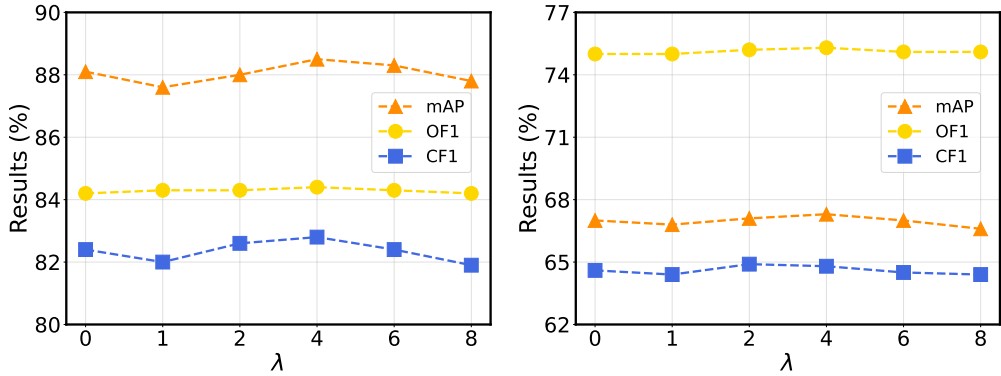

Figure 6: **Sensitivity analysis (%) of $\lambda$.** Results are reported on MS-COCO (left) and NUS-WIDE (right).

follow the official data split: 70% of the images are used for training, 10% for validation, and 20% for testing. The detailed per-class results are presented in Table 14. Our method (constructed on ResNet-101) achieves a state-of-the-art mean score of 86.1%, surpassing recent competitive methods such as GCF-Net and Anatomy-XNet. Specifically, MVLR secures the highest performance in the majority of the 14 thoracic disease categories, underscoring its exceptional generalization capability and robustness in specialized, real-world applications.

Table 13: **Performance comparison.** Results are reported on MultiScene (Hua et al., 2021) and MLRSNet (Qi et al., 2020).

| Method | Backbone | MultiScene | | | | | MLRSNet | | | | |
| | | mAP | ALL | | Top-3 | | mAP | ALL | | Top-3 | |
| | | | CF1 | OF1 | CF1 | OF1 | | CF1 | OF1 | CF1 | OF1 |
|---|---|---|---|---|---|---|---|---|---|---|---|
| RemoteCLIP-FT | ResNet50 | 66.9 | 63.2 | 73.9 | 52.0 | 65.2 | 97.7 | 90.6 | 92.6 | 67.9 | 70.1 |
| **MVLR** | ResNet50 | 67.9 | 64.8 | 75.4 | 52.3 | 66.3 | 98.0 | 91.2 | 92.8 | 66.9 | 70.6 |
| RemoteCLIP-FT | ViT-B/32 | 67.2 | 63.3 | 73.7 | 52.5 | 65.6 | 97.8 | 91.1 | 93.1 | 66.7 | 70.2 |
| **MVLR** | ViT-B/32 | 68.1 | 65.3 | 75.2 | 53.5 | 66.3 | 98.4 | 92.4 | 93.7 | 67.2 | 70.2 |

Table 14: Performance comparison of different methods across various categories on ChestX-ray14 (Wang et al., 2017).

| Method | Ate | Car | Eff | Inf | Mass | Nod | Pna | Pnx | Con | Ede | Emp | Fib | Pt | Her | Mean |
|---|---|---|---|---|---|---|---|---|---|---|---|---|---|---|---|
| CheXNet (Rajpurkar et al., 2017) | 78.0 | 82.2 | 82.7 | 68.9 | 83.1 | 78.1 | 73.5 | 85.1 | 75.4 | 85.0 | 92.5 | 82.2 | 79.3 | 93.2 | 81.8 |
| LAAGNet (Chen et al., 2019a) | 78.3 | 88.5 | 83.4 | 70.3 | 84.1 | 79.0 | 72.9 | 87.7 | 75.4 | 85.1 | 93.9 | 83.2 | 79.8 | 91.6 | 82.4 |
| CheXGCN (Chen et al., 2020) | 78.6 | 89.3 | 83.2 | 69.9 | 84.0 | 80.0 | 73.9 | 87.6 | 75.1 | 85.0 | **94.4** | 83.4 | 79.5 | 92.9 | 82.6 |
| ADNet (Kang et al., 2024) | 80.1 | 89.1 | 84.3 | **71.7** | 84.9 | **82.2** | 75.0 | 89.7 | 76.9 | 86.5 | 94.2 | 95.3 | 80.8 | 96.2 | 83.8 |
| Anatomy-XNet (Kamal et al., 2022) | 83.1 | **91.4** | 88.6 | **71.7** | 86.0 | 80.4 | 77.1 | 88.2 | 80.9 | 89.9 | 92.9 | 84.4 | 79.8 | **96.4** | 85.1 |
| GCF-Net (Sun et al., 2026) | 83.6 | 89.6 | 88.5 | 70.8 | 88.4 | 80.7 | 79.5 | 90.6 | 82.5 | 91.1 | 92.8 | 83.2 | 82.8 | 95.0 | 85.7 |
| Ours | **84.1** | 90.1 | **89.0** | 71.0 | **88.9** | 81.3 | **79.8** | **91.0** | **83.0** | **91.5** | 93.3 | **83.6** | **83.5** | 95.3 | **86.1** |

**Experiments on different pre-trained vision-language models.** To explore the potential influence of pre-training data and noise, we evaluate our method using various pre-trained vision-language models, including MetaCLIP2 (trained on Meta's proprietary data), SigLIP2 (trained on Google's data), and OpenCLIP (trained on public datasets). For a fair comparison, all models are based on the ViT-B/16 backbone. As shown in Table 15, while noisy pre-training data slightly degrades OpenCLIP's performance, it still achieves competitive results, demonstrating the robustness of the proposed modules. Moreover, the improvements from SigLIP2 primarily stem from its pairwise sigmoid loss instead of a traditional softmax loss, which is inherently more suited for multi-label task.

Table 15: Performance comparison of different pre-trained vision-language models on MS-COCO and NUS-WIDE datasets. All models use ViT-B/16 as the backbone.

| Method | MS-COCO | | | | | NUS-WIDE | | | | |
| | mAP | ALL | | Top-3 | | mAP | ALL | | Top-3 | |
| | | CF1 | OF1 | CF1 | OF1 | | CF1 | OF1 | CF1 | OF1 |
|--------|------|------|------|------|------|------|------|------|------|------|
| OpenCLIP | 89.8 | 84.4 | 85.5 | 78.9 | 80.2 | 68.5 | 65.5 | 75.6 | 60.6 | 70.5 |
| CLIP | 90.4 | 84.8 | 86.2 | 79.7 | 81.2 | 68.9 | 66.1 | 76.1 | 61.7 | 71.7 |
| MetaCLIP2 | 90.7 | 85.0 | 86.7 | 79.8 | 80.8 | 69.0 | 66.4 | 76.1 | 61.9 | 71.4 |
| SigLIP2 | **91.1** | **85.6** | **87.0** | **80.9** | **82.0** | **69.5** | **67.3** | **77.1** | **62.4** | **72.1** |

**Qualitative results.** In Figure 7, we provide the predicted results of "Classifier Learning" approach and our proposed MVLR with a threshold of 0.6. **1)** Baseline generates many false negatives, especially the small objects in complex scenarios. **2)** Baseline tends to predict "person" in most cases. We attribute this to the fact that the static category centers in the Baseline tend to overfit the distribution of the training set, resulting in high confidence in the frequently occurred labels such as "person". In contrast, dynamically constructing the input-adaptive category centers (i.e., in MVLR) mitigates this issue.

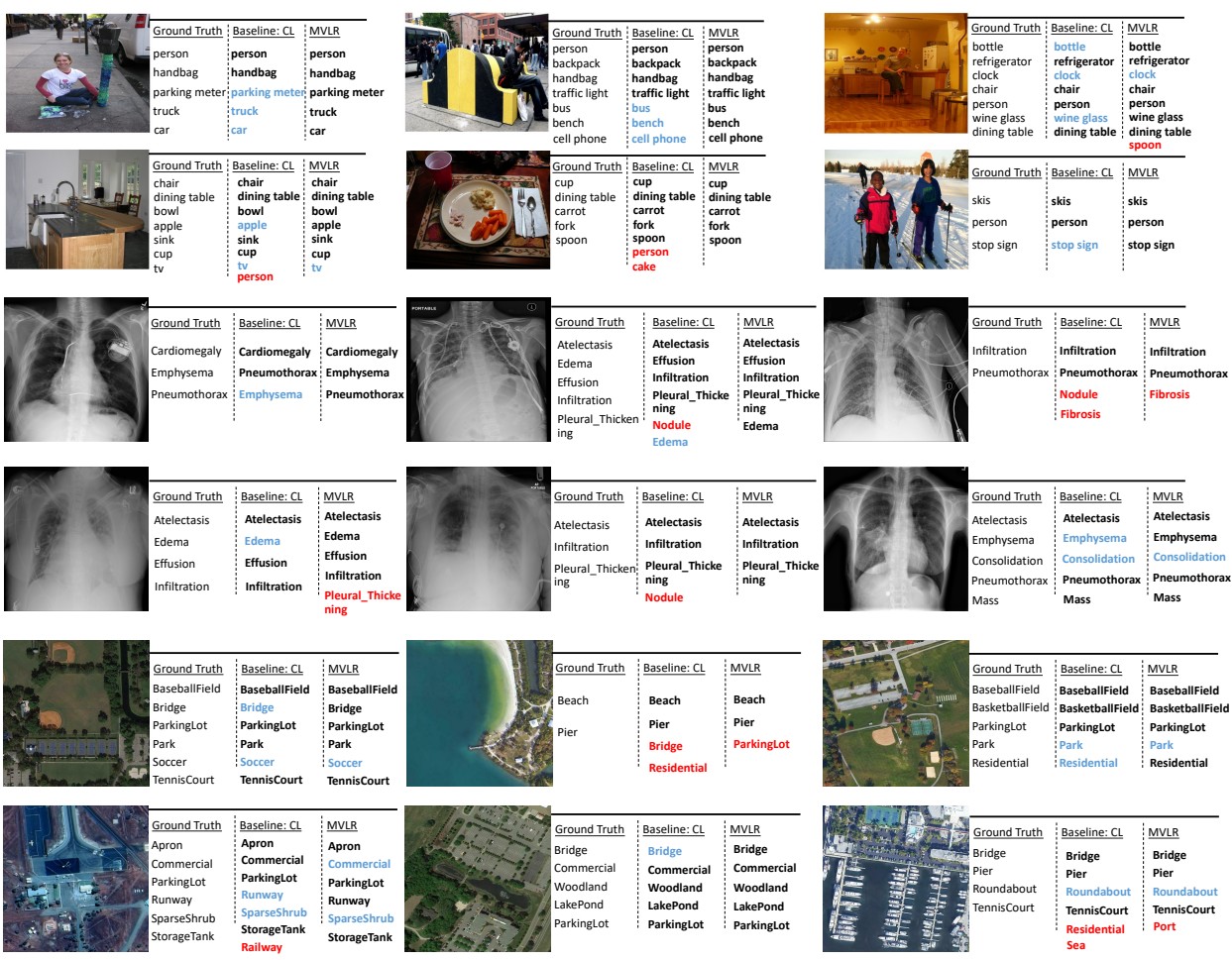

Figure 7: **Predictions of "Classifier Learning" (CL) baseline and our proposed MVLR.** The provided samples showcase diverse scenarios including natural, remote sensing, and medical images. Each sample is marked with three columns. From the left to right are ground truth labels, predictions of baseline and predictions of MVLR, respectively. Correct predictions are marked in bold font. Missed objects are highlighted in blue and incorrect predictions are marked in red. Best viewed in colors.

