# OpenReview forum: "Mamba-Enhanced Visual-Linguistic Representation for Multi-Label Image Recognition"
_TMLR — Accepted by TMLR_

### Review · Reviewer_WM6Y · 2025-12-29

**Summary Of Contributions:**

This paper proposes MVLR, a Mamba‑enhanced visual–linguistic representation framework for multi-label image recognition. The method consists of three major components:

PDLR: A prompt-driven module that extracts label representations using hard and soft prompts.
IFM: An Interaction and Fusion Module that combines label relations (via label attention), context-aware label features (via cross-attention), and channel interaction with a regularization term.
QMVL: A Quadruplet Mamba-based cross-modal interaction block designed to model bidirectional relations between visual features and linguistic embeddings.

The authors report state-of-the-art performance on MS‑COCO, PASCAL VOC 2007, and NUS‑WIDE, along with ablations.

**Additional Comments:**

QMVL motivation: What unique interactions are captured by each of H1/H2/H3/H4? Why is partial reversal theoretically better than overall reversal?

Scanning vs. overhead: Since SSM blocks inherently scan, please disentangle QMVL’s cost from base SSM scanning and report latency/memory deltas.

Controlled baselines: Can you provide no‑Mamba and single‑pass Mamba fusion baselines at matched compute?

Comparative context: Please add discussion/experiments referencing SiMBA situate MVLR among recent SSM + spectral/channel‑mixing alternatives.

**Audience:**

Yes

**Audience Explanation:**

Yes, but with reservations.
TMLR’s audience—researchers working on multimodal learning, vision–language models, SSMs/Mamba variants—would likely find the idea of a Mamba‑based cross‑modal fusion module interesting. The topics of prompt‑driven label embeddings and label‑context fusion (IFM) are also relevant.
However, the impact is reduced by:

Limited comparison to relevant state‑of‑the‑art multimodal architectures.
Lack of positioning relative to recent SSM-based models (e.g., ViM, LocalMamba, MambaVision, SiMBA, VMamba, Mamba‑360 survey insights).
Architectural complexity that obscures the core contribution.

**Broader Impact Concerns:**

QMVL Ablation Is Not Fully Controlled
The ablations compare different scanning orders but lack:

Removing Mamba entirely (e.g., replacing QMVL with regular self-attention)
Comparing QMVL to more powerful cross-modal attention architectures
Ablations on scanning order are helpful, but need to add:

A no‑Mamba baseline (self‑attention or cross‑attention only) at matched compute.
A single‑pass Mamba fusion (no quadruplet) vs. QMVL at matched FLOPs/params.
A module‑wise cost–benefit chart (accuracy vs. compute for PDLR/IFM/QMVL).

This makes it unclear whether performance gains come from Mamba or simply from increased fusion depth.

**Claims And Evidence:**

No

**Claims Explanation:**

Partially supported.
While the paper provides extensive empirical results on three benchmarks and offers ablation studies for its modules (PDLR, IFM, QMVL), several core claims are not fully substantiated. In particular:

The motivation for the quadruplet Mamba (QMVL) design is unclear, and the paper does not provide theoretical justification for the H1/H2/H3/H4 constructions or why Mamba is the appropriate mechanism for cross‑modal fusion.
Claims about “fully exploiting linguistic modality,” “bidirectional interaction,” and “order‑independent modeling” lack rigorous analysis or evidence.
Efficiency claims are only supported by FLOPs tables; no latency, memory, or runtime evaluation is provided.
The generalization claim is overstated because experiments are limited to older datasets (COCO, VOC, NUS‑WIDE).
Comparisons are missing with strong multimodal baselines (e.g., ALBEF, BLIP, Flamingo) and recent SSMs (VIM, LocalMamba, MambaVision, SiMBA, VMamba, Mamba‑360 survey insights).

Thus, while there is good empirical evidence for performance gains, some key scientific claims remain insufficiently supported.

**Requested Changes:**

Major Weaknesses (Core Issues)

1. Methodological Complexity Without Clear Justification:

The proposed pipeline combines five interacting components (hard prompts, soft prompts, two types of attention, channel interaction, regularization, quadruplet scanning with Mamba). The paper does not clearly explain why each component is necessary, how they complement each other, or whether a simpler fusion would suffice. Several modules appear heuristically combined rather than theoretically motivated, which raises concerns about conceptual clarity and over-engineering.

2. Insufficient Theoretical Motivation for Mamba:

While Mamba is highlighted as a strength, the paper provides no clear theoretical grounding or intuition for why Mamba Mamba (vs. standard attention) is preferable for cross‑modal fusion in multi‑label classification. Specific concern about QMVL: the introduction of H1/H2/H3/H4 concatenations (normal/inverse orders across modalities) appears ad‑hoc. The paper should articulate what each concatenation captures, why partial reversal outperforms overall reversal, and how it reduces order dependence beyond empirical observation. Right now, this looks like “using Mamba for its own sake” rather than motivated architectural design. The quadruplet scanning mechanism also seems empirically motivated, without explanation of why forward–reverse concatenation improves cross-modal interaction.

Actionable request: Provide a formal or semi‑formal analysis (or at least a clear intuition) for the quadruplet construction:

What interactions are uniquely captured by H1/H2 vs. H3/H4?
Why is partial reversal (per modality) theoretically beneficial for cross‑modal alignment?
How does the QMVL block relate to standard bidirectional or registry‑augmented SSM designs?

3) Scanning vs. Overhead: Clarify What QMVL Actually Adds:

SSM/Mamba layers already involve scanning; thus, QMVL may not be an “extra overhead” block so much as a specific way of ordering and fusing sequences. The manuscript should explicitly clarify that QMVL’s cost comes from the quadruplet concatenations and dual Mamba passes, and disentangle the inherent SSM scanning from the design choices (H1–H4).
Please add a measured breakdown: FLOPs/params/latency for each QMVL variant relative to a single Mamba block and to self‑/cross‑attention fusion.


4. Weak Positioning in Related Work:

The related work section is incomplete or outdated:

No comparisons to strong VLM-based fusion architectures (e.g., BLIP, BLIP‑2, ALBEF, Flamingo).
No discussion of label correlation modeling literature beyond basic GCN/RNN/Transformer methods.
No mention of prompt tuning works like CoOp, CoCoOp, Tip-Adapter, etc., despite heavy reliance on prompts.

This weakens claims of novelty.


5) Missing Comparisons to Modern SSM Baselines:

Comparisons against recent SSM vision/multimodal works are thin. In particular, a comparative discussion with VIM/LocalMamba/MambaVision/SiMBA (a simplified Mamba + spectral channel mixing architecture) would be useful for readers:

Actionable request: Add a comparative paragraph referencing recent VIM/LocalMamba/MambaVision/SiMBA variants, clarifying how MVLR/QMVL differs in motivation, stability, and fusion cost. Although MVLR is compared with some classical approaches, the evaluation lacks:
Transformer‑based multimodal baselines (e.g., ALBEF, METER, Pixel-BERT)
Recent Mamba‑based multimodal models
CLIP-based multi-label adapters

As a result, the paper does not convincingly establish that its improvements hold against true SOTA multimodal methods.

6) Limited Dataset Scope & Generalization:

Evaluation is restricted to MS‑COCO, VOC‑2007, and NUS‑WIDE (older, smaller benchmarks). For a cross‑modal method, broader validation on Visual Genome, Open Images v6, LVIS (multi‑label), or LAION‑derived subsets would better support claims of robust generalization.

7. Computational Efficiency Claims Are Incomplete:

FLOPs are reported, but latency, GPU memory, training time per epoch, and scalability beyond ResNet‑101/ViT‑B/16 are missing. QMVL’s efficiency claim should be supported by wall‑clock and memory profiles.
Inference latency
GPU memory usage
Training time per epoch
Scalability beyond ResNet101 and ViT-B/16

Thus, efficiency claims remain only partially substantiated.


8. Writing & Presentation Issues:

Long and dense textual descriptions make some sections difficult to follow.
Several figures (especially Fig. 1 & Fig. 3) are visually cluttered.
Some sections are dense, and figures (especially the QMVL schematic) are visually cluttered. Streamline the motivation → design → cost → gain narrative for QMVL, and rename/explain H1–H4 to reflect their role (e.g., “V→L / L→V / reverse‑V / reverse‑L”) with explicit intent and expected effect.
Some claims seem overstated (“fully exploit linguistic modality”, “first attempt”), given incomplete comparisons.

---

> ### Author Response · Authors · 2026-02-15
> **Rebuttal for Reviewer WM6Y**
>
> ### Q1. Methodological Complexity Without Clear Justification
> We respectfully clarify that our framework is not a heuristic combination, but a structured design where each component addresses a specific limitation in multi-label image recognition. All proposed components are functionally orthogonal and the motivation has been  clarified in the introduction. To be specific,  PDLR extracts well linguistic embeddings for all labels, IFM explores the label relations among the extracting label representations and then aggregates the relation-enhanced label representations together, and QMVL conducts a mutual interaction between visual and linguistic modalities. Moreover, While we rely on empirical validation rather than theoretical derivation, our ablation studies (Section 4.4, Table 4\5\6\7\8) demonstrate that the components are not redundant; they are functionally orthogonal and collectively contribute to the final performance.
>
> ### Q2. Insufficient Theoretical Motivation for Mamba
> We clarify that QMVL is not an ad-hoc combination, but a necessary mechanism to approximate global attention within a causal, linear-complexity framework ($O(N)$).
>
> **Why Mamba?** Standard Attention ($O(N^2)$) is too costly for long multi-modal sequences. Mamba is efficient ($O(N)$) but strictly unidirectional. In a sequence $[A, B]$, modality $B$ sees $A$, but $A$ cannot see $B$. This creates a unidirectional interaction.
>
> **Theoretical Logic of H1-H4.** QMVL constructs a "virtual" global receptive field by decomposing interaction into four paths:
> - H1 ($A \to B$) & H3 ($B \to A$): By physically swapping modality order, we force the causal SSM to model both $P(B|A)$ and $P(A|B)$. This solves the bottleneck, ensuring bidirectional cross-modal conditioning.
> - H1 vs. H2, H3 vs. H4 (reversed scans): We partially reverss the order of modalities, which eliminates the dependency on order to some extent.
>
> **Why Partial Reversal?** Overall Reversal just changes scan direction.Partial Reversal (reversing $A$ and $B$ individually) helps to align the semantic endpoints of modalities.
>
> ### Q3. Scanning vs. Overhead: Clarify What QMVL Actually Adds
> We have added a comprehensive comparison of FLOPs, Parameters, and Inference Latency against both the baseline and SOTA methods, which demonstrates our method's superior efficiency-accuracy trade-off (see Sec. A.2 and Table 9, also see the responses to Q7). Regarding the comparison with Self-/Cross-Attention fusion, we respectfully note that this was already presented in Table 6 of our original submission, confirming QMVL's significant advantage over traditional attention mechanisms. We believe that we have fully validated the effectiveness and reliability of our design without requiring further granular decomposition.
>
> ### Q4. Weak Positioning in Related Work
> We thank the reviewer and will incorporate these references to clarify our contribution. By the way, we respectfully clarify that CoOp and CoCoOp are cited in Section 3.2 Soft prompts. We have modified the related work section with discussing more recent works in our revised manuscript.

---

> ### Author Response · Authors · 2026-02-15
> **Rebuttal for Reviewer WM6Y**
>
> ### Q5. Comparisons to Modern SSM Baselines
>
> Thanks for the comments. We re-implemented MambaVision-B and SiMBA-Base on MS-COCO and NUS-WIDE. As presented in Tables 5 and 6, these SSM baselines consistently underperform compared to our MVLR. Specifically, MVLR outperforms SiMBA by **6.2% mAP** and **5.2% OF1** on the MS-COCO dataset, and by **5.4% mAP** and **3.3% OF1** on the NUS-WIDE dataset. Furthermore, our method maintains a clear advantage over MambaVision, demonstrating the superior effectiveness of MVLR over existing SSM baselines. The analysis has been added to our revised manuscript. Details can be founded in the Sec. A2 in our revised manuscript.
>
> - The comparisons on the MS-COCO dataset.
>
> | Method         | mAP      | ALL  |          |          |      |          |          | Top-3 |          |          |      |          |          |
> | -------------- | -------- | ---- | -------- | -------- | ---- | -------- | -------- | ----- | -------- | -------- | ---- | -------- | -------- |
> |                |          | CP   | CR       | CF1      | OP   | OR       | OF1      | CP    | CR       | CF1      | OP   | OR       | OF1      |
> | MambaVision-B  | 86.9     | 86.2 | 74.7     | 80.0     | 87.4 | 76.3     | 81.5     | 90.3  | 66.9     | 76.8     | 91.6 | 68.2     | 78.2     |
> | SiMBA-Base     | 82.3     | 82.5 | 73.6     | 77.8     | 83.3 | 75.5     | 79.2     | 88.3  | 64.7     | 74.7     | 89.2 | 66.4     | 76.1     |
> | **MVLR**-rn101 | **88.5** | 83.1 | **82.5** | **82.8** | 83.5 | **85.3** | **84.4** | 88.7  | **69.4** | **77.9** | 90.5 | **71.3** | **79.8** |
>
> - The comparisons on the NUS-WIDE dataset.
>
> | Method          | mAP      | ALL      |          | Top-3    |          |
> | --------------- | -------- | -------- | -------- | -------- | -------- |
> |                 |          | CF1      | OF1      | CF1      | OF1      |
> | MambaVision-B   | 65.8     | 64.6     | 74.7     | 59.7     | 70.4     |
> | SiMBA-Base      | 61.9     | 60.3     | 72.2     | 56.0     | 68.8     |
> | MVLR-rn101      | **67.3** | **64.9** | **75.5** | **60.0** | **71.5** |
>
>
> ### Q6. Limited Dataset Scope & Generalization
> We respectfully clarify that MS-COCO, VOC-2007, and NUS-WIDE are the de facto standard benchmarks for the multi-label image recognition task, and this is the community consensus where all recent SOTA methods (e.g., Q2L, ADD-GCN, TresNet, MambaML) benchmark exclusively on these datasets. The datasets suggested (LVIS, OpenImages) are typically used for Long-Tail Recognition or Detection. Adopting them would prevent direct, fair comparison with existing literature, as standard multi-label image recognition baselines do not report results on them.
>
>
> ### Q7. Computational Efficiency
>
> Thanks for the comments. We have provided additional efficiency comparisons, including FLOPs, number of parameters, inference latency, and peak GPU memory consumption. As demonstrated in Table 4, our method achieves superior performance with highly efficient FLOPs and inference latency, and the inference latency is significantly lower than that of Q2L and SiMBA. Specically, (1) the entire framework requires only **13.1ms** to process a single image with a peak memory footprint of just 677.2MB, validating the overall efficiency of our framework. (2) Compared to the baseline, our additional modules do not increase peak memory usage and significantly boost recognition performance at the cost of only a slight increase in latency, further proving the effectiveness and efficiency of the proposed components.
> Details can be founded in the Sec. A2 in our revised manuscript.
>
> |                | FLOPs (G) | Params. (M) | Latency (ms) | Peak Mem. (MB) | COCO | NUS  |
> | -------------- | --------- | ----------- | ------------ | -------------- | ---- | ---- |
> |                |           |             |              |                |      |      |
> | ML-Decoder     | 37.2      | 47.3        | 14.8         | 357.9          | 86.6 | 64.2 |
> | Q2L            | 43.2      | 143.1       | 18.0         | 2369.3         | 84.9 | 65.0 |
> | MambaVision    | 14.9      | 97.6        | 9.9          | 391.1          | 86.9 | 65.8 |
> | SiMBA          | 39.1      | 62.5        | 21.4         | 336.3          | 82.3 | 61.9 |
> | Baseline       | 36.8      | 43.6        | 9.1          | 677.0          | 81.6 | 58.6 |
> | **MVLR** | 37.6      | 47.8        | 13.1         | 677.2          | 88.5 | 67.3 |

---

> ### Author Response · Authors · 2026-02-15
> **Rebuttal for Reviewer WM6Y**
>
> ### Q8. Writing & Presentation Issues
> We respectfully decided to retain the notation $H_1$--$H_4$ for the following reason, where the relationships modeled by these hypotheses go beyond simple directionality (e.g., V$\to$L). Using simplified labels like "reverse-V" would be imprecise and fail to capture the actual logic behind them. Accordingly, we maintain the original notation while strengthening the textual explanations associated with  $H_1$--$H_4$. Moreover, we will restructure the QMVL section to strictly follow the "Motivation $\to$ Design $\to$ Gain" logic (see our new manuscript). We will tone down superlatives (e.g., "fully exploit", "first attempt") to ensure objective, evidence-based phrasing.
>
> ### Q9. QMVL Ablation Is Not Fully Controlled
> We respectfully note that we have conducted these controlled experiments, which confirm the effectiveness of our design. To be specific, the experiments of replacing QMVL with regular self-attention have shown in Table 6, where the classic self-attention and cross-attention are taken for comparisons. Not only are MVLR achieves better performance, but it also has much fewer FLOPs. Moreover, the ablation studies on all components (i.e., PDLR, IFM and QMVL) have been clearly investigated, and the performance gains of these modules have been clearly verified.
>
> ### Q10. Controlled baselines
> The experiments of no-Mamba baseline can be founded in Table 6, where the Mamba structure is replaced with the classic attention modules. The experiments of the single-pass Mamba can be founded in Table 5 (denoted as 'F.').

---

### Review · Reviewer_kinT · 2026-01-07

**Summary Of Contributions:**

This paper proposes a new framework termed MVLR (Mamba-enhanced Visual-Linguistic Representation) for multi-label image recognition, which features fully use of language models and visual–textual interactions.

The key technical strengths are:
- A Prompt-Driven Label Representations learning that develops hard prompts and soft prompts to build label embeddings, capturing complementary aspects of label semantics.
- A Interaction and Fusion Module that deeply aggregates the captured label representations, yielding final relation-enhanced label representations.
- A Quadruplet Mamba-Enhanced Visual-Linguistic block that leverages state space models for a mutual interaction between visual and linguistic modalities.

The key strengths are:
- The proposed MVLR framework is well-motivated and each key component is clearly articulated.
- The introduction of state space models for modeling the relationship between label representation and visual feature sequences is a novel approach in the field of multi-label image classification.
- Using learned label embeddings as a dynamic classifier instead of a fixed linear head is impressive, and ablation experiments have confirmed the effectiveness of this idea.
- The overall framework has a significant performance advantage over existing methods, and the ablation experiments are also very thorough.

The key weaknesses are:
- The writing need be improved and many typos should be corrected, such as "representthe" in the last line on page 2.
- The complexity of the entire framework is concerning. During training, the text encoder has to perform a forward pass at each step because the soft prompts need to be constantly updated, which obviously increases the training burden.
- The motivation of the knowledge-to-context regularization (KCR) loss is unclear. Intuitively, it seems more beneficial for model performance to keep $T^{ca}$ and $T^{ka}$ as inconsistent as possible. If we constrain them to be as consistent as possible using KCR loss, it seems meaningless to concatenate them as input to subsequent modules; we can simply choose one of them arbitrarily.

**Additional Comments:**

Nan

**Audience:**

Yes

**Audience Explanation:**

1. Topical fit: It addresses multi-label image recognition with modern vision–language models (CLIP, prompt tuning) and state space models / Mamba, all of which are active topics in the ML community and relevant to TMLR.
2. Methodological interest: Combining hard and soft prompts for label embeddings, a dedicated label interaction module, and a Mamba-based cross-modal fusion block are all techniques that will appeal to researchers working on representation learning, multimodal learning, and efficient sequence models.
3. Empirical contribution: The paper reports consistent SOTA results on standard multi-label benchmarks (MS-COCO, Pascal VOC, NUS-WIDE), which is relevant to practitioners and benchmark-oriented researchers.

**Broader Impact Concerns:**

No concerns related to work ethics impact

**Claims And Evidence:**

No

**Claims Explanation:**

1. In the introduction section, the authors mention that existing methods overlook the hierarchical and compositional nature of semantic structures among labels, and therefore propose the Interaction and Fusion Module. However, in subsequent chapters, the reviewers found no discussion on this topic, and only the performance improvement in ablation experiments is unconvincing. The authors are encouraged to explain how the proposed module captures the hierarchical and combinatorial characteristics of label semantics.

**Requested Changes:**

1. The key technical aspects of visual feature extraction in this work are not clearly described. Based on the lack of detailed implementation information, it appears the authors used the CLIP visual encoder to extract image features, which is clearly unfair compared to current methods whose visual encoders are pre-trained on ImageNet. The authors should clarify this point and provide a fairer comparison. (major)
2. Regarding the use of state space models for multi-label image classification, the authors are encouraged to compare their methods with MambaML [1], thereby providing readers with a deeper understanding of Mamba's role in multi-label tasks. (minor)

[1] MambaML: Exploring State Space Models for Multi-Label Image Classification. ICCV 2025.

---

> ### Author Response · Authors · 2026-02-15
> **Rebuttal for Reviewer kinT**
>
> ### Q1. The writing need be improved and many typos should be corrected.
> Thank you for your comments. We have thoroughly proofread the entire manuscript to correct grammatical errors and improve the flow of the writing, like "representthe" -> "represent the".
>
> ### Q2. The complexity of the entire framework.
>
> Thanks for the comments. We have provided additional efficiency comparisons, including FLOPs, number of parameters, inference latency, and peak GPU memory consumption. As demonstrated in Table 4, our method achieves superior performance with highly efficient FLOPs and inference latency, and the inference latency is significantly lower than that of Q2L and SiMBA. Specically, (1) the entire framework requires only **13.1ms** to process a single image with a peak memory footprint of just 677.2MB, validating the overall efficiency of our framework. (2) Compared to the baseline, our additional modules do not increase peak memory usage and significantly boost recognition performance at the cost of only a slight increase in latency, further proving the effectiveness and efficiency of the proposed components.
>
> |                | FLOPs (G) | Params. (M) | Latency (ms) | Peak Mem. (MB) | COCO | NUS  |
> | -------------- | --------- | ----------- | ------------ | -------------- | ---- | ---- |
> |                |           |             |              |                |      |      |
> | ML-Decoder     | 37.2      | 47.3        | 14.8         | 357.9          | 86.6 | 64.2 |
> | Q2L            | 43.2      | 143.1       | 18.0         | 2369.3         | 84.9 | 65.0 |
> | MambaVision    | 14.9      | 97.6        | 9.9          | 391.1          | 86.9 | 65.8 |
> | SiMBA          | 39.1      | 62.5        | 21.4         | 336.3          | 82.3 | 61.9 |
> | Baseline       | 36.8      | 43.6        | 9.1          | 677.0          | 81.6 | 58.6 |
> | **MVLR**-RN101 | 37.6      | 47.8        | 13.1         | 677.2          | 88.5 | 67.3 |
>
>
> ### Q3. The motivation of the KCR loss is unclear.
> We understand the intuition that "distinct features provide more information," but we respectfully clarify that the goal of KCR is semantic alignment, not redundancy. Our motivation comes from the following aspects:
> - **Diverse values of $T^{ca}$ and $T^{ka}$ lead to the misalignment of visual-linguistic features**
> The primary goal of our proposed MVLR is to capture aligned visual-linguistic representations for multi-label image recognition. While diversity is important, unconstrained diversity in the linguistic space can be detrimental when aligning with visual features. Excessive divergence between these two representations （$T^{ca}$ and $T^{ka}$）not only burdens the subsequent fusion module but also destabilizes the later alignment with visual features.
>
> - **Consistency loss $\neq$ Identical features**
> It is crucial to clarify that the KCR loss—modulated by a specific loss coefficient—does not force $T^{ca}$ and $T^{ka}$ to be mathematically identical. Actually, the two features still retain complementary details and variations of different sources of information.
>
> - **Robustness**
> By constraining these two representations to be consistent, the KCR loss essentially performs a form of mutual learning or self-distillation. The consistency constraint ensures that if one branch captures a noisy or outlier feature, the signal from the other branch helps correct it.
>
> Therefore, KCR strikes a balance: it prevents the two branches from drifting apart while allowing them contain complementary information, which facilitate the network to achieve better performance (see Fig .6, obtain 0.2% mAP performance gains).

---

> ### Author Response · Authors · 2026-02-15
> **Rebuttal for Reviewer kinT**
>
> ### Q4. The key technical aspects of visual feature extraction... it appears the authors used the CLIP visual encoder...
> We appreciate the reviewer’s scrutiny regarding the choice of the visual backbone and the fairness of the comparison. We address this concern from three perspectives:
>
> - **Fair comparisons in ablation studies**
> Although we have employ the CLIP visual encoder to extract image features, the effectiveness of all our proposed core components have been verified in the ablation studies (as detailed in Table 4-5). The results demonstrate that every individual module provides a distinct and significant performance gain. For example, the proposed PDLR, IFM and QMVL clearly improves the mAP performance from 81.8\% to 88.5\% based the basedline method, and all experiments are fairly conducted. This confirms that the improvements stem from our proposed modules rather than simply relying on the strength of the backbone itself.
>
> - **Fair comparison with SOTA of CLIP backbone (e.g., PatchCT)**
> The unfair comparison concern raised by the authors is unfounded. We did not use the CLIP backbone while only comparing it with backbones pre-trained on ImageNet; on the contrary, we also conducted comparisons with methods adopting similar CLIP architecture across all three datasets, and our approach demonstrated distinct advantages on all three datasets. To be specific, the recent method PatchCT is also build upon the CLIP architecture for multi-label classification, we have fairly compared the proposed method with PatchCT with the same backbone ( ViT-ViT-B/16) on MS-COCO, Pascal VOC 2007 and NUS-WIDE as shown in Table 1-3, and our method consistently achieves superior performance. For example, our proposed method outperforms PatchCT by 2.1% on mAP accuracy of MS-COCO dataset.
>
>
> - **Methodological necessity (vision-language alignment)**
> Our proposed method is explicitly designed to exploit the intrinsic semantic correlations between visual and textual modalities. Therefore, our method requires aligned visual and linguistic features simultaneously, which is different from some previous methods that rely solely on visual features. This is also the main reason why we utilize the CLIP pre-trained backbone rather than the backbone pretrained on ImageNet. Although we did not conduct experiments using an ImageNet pre-trained backbone, the two points explained above—namely, the fair comparison in our ablation studies and the comparison against PatchCT—still fully demonstrate the effectiveness of our approach.
>
> We have revised Section 4 Experiments in the manuscript to explicitly clarify the rationale for using CLIP and to highlight the direct comparison with PatchCT to clarify the fair comparison on CLIP encoders.

---

> ### Author Response · Authors · 2026-02-15
> **Rebuttal for Reviewer kinT**
>
> ### Q5. Comparison of the proposed method with MambaML.
> We thank the reviewer for pointing out this relevant recent work. We have added a comparison with MambaML in the revised manuscript (as shown in Table 1-3). As summarized below, our method consistently outperforms MambaML across the MS-COCO, NUS-WIDE, and Pascal VOC 2007 datasets. For example, our MVLR achieves mAP values of 88.5% (448×448) and 89.0% (576×576), both surpassing MambaML’s respective scores of 85.7% and 86.7%. The detailed comparisons of the three datasets are listed in the following tables (also cound be founded in our new manuscript).
>
> - The comparisons on the Microsoft COCO dataset
>
> | Method       | Backbone   | Resolution | mAP    | ALL              |              |              |              |              |              | Top-3            |              |              |              |              |              |
> |--------------|------------|------------|--------|------------------|--------------|--------------|--------------|--------------|--------------|------------------|--------------|--------------|--------------|--------------|--------------|
> |              |            |            |        | CP               | CR           | CF1          | OP           | OR           | OF1          | CP               | CR           | CF1          | OP           | OR           | OF1          |
> | MambaML (ICCV'25)       | ResNet101  | (448, 448) | 85.7   | **86.1**             | 75.1         | 80.3         | **87.7**         | 77.6         | 82.3         |**89.4**             | 66.5         | 76.3         | 91.5         | 68.2         | 78.1         |
> | **MVLR**     | ResNet101  | (448, 448) | **88.5** | 83.1             | **82.5**     | **82.8**     | 83.5         | **85.3**     | **84.4**     | 88.7             | **69.4**     | **77.9**     | 90.5         | **71.3**     | **79.8**     |
> | MambaML (ICCV'25)       | ResNet101  | (576, 576) | 86.7   | **86.9**             | 76.4         | 81.3         | **88.0**         | 79.0         | 83.2         | **90.0**             | 67.1         | 76.9         | 91.9         | 68.8         | 78.7         |
> | **MVLR**     | ResNet101  | (576, 576) | **89.0** | 83.0             | **83.7**     | **83.3**     | 83.7         | **86.8**     | **85.2**     | 89.3             | **70.2**     | **78.6**     | 91.5         | **72.0**     | **80.6**     |
>
> - The comparisons on the NUS-WIDE dataset.
>
> | Method      | Backbone | mAP    | ALL      |         | Top-3    |         |
> |--------------|--------|--------|----------|---------|----------|---------|
> |            |    |        | CF1      | OF1     | CF1      | OF1     |
> | MambaML (ICCV'25)    | ResNet101   | 65.9   | 63.7     | 75.0    | 59.8     | 70.7    |
> | **MVLR**   | ResNet101  | **67.3** | **64.9** | **75.5** | **60.0** | **71.5**    |
>
>
> - The comparisons on the Pascal VOC 2007 dataset
>
> | Method  | Backbone | aero  | bike  | bird  | boat  | bottle | bus   | car   | cat   | chair | cow   | table | dog   | horse | motor | person | plant | sheep | sofa  | train | tv    | mAP   |
> |--------------|--------|-------|-------|-------|-------|--------|-------|-------|-------|-------|-------|-------|-------|-------|-------|--------|-------|-------|-------|-------|-------|-------|
> | MambaML (ICCV'25)  | ResNet101     | **99.8**  | **98.6**  | 97.8  | 98.0  | **82.8**| 96.3  | 98.1  | 98.3  | 84.0  | 96.7  | 88.3  | 98.2  | 98.6  | 96.8  | 99.0   | 87.5  | 96.8  | **89.8**  | 99.2  | **95.1**  | 95.0  |
> | **MVLR**  | ResNet101     | 99.7  | 98.1  | **98.5**  | **99.3**| 87.0  | **98.2**| **98.3**| **98.9**| **86.7**| **98.3**| **89.9**| **99.2**| **98.7**| **97.7**| **99.3**   | **88.3**| **97.6**| 87.0  | **99.3**  | 94.1  | **95.7**|

---

> > ### Comment · Reviewer_kinT · 2026-02-26
> > **A thorough response and the doubts were addressed.**
> >
> > Thank you for the author's reply. My concern has been resolved.

---

### Review · Reviewer_QwaN · 2026-03-11

**Summary Of Contributions:**

This paper proposes a Mamba-Enhanced Visual-Linguistic Representation (MVLR) framework for multi-label image recognition, addressing the underutilization of linguistic modalities in existing vision-language methods. Key contributions are: 1) Designing Prompt-Driven Label Representation (PDLR) with hard and soft prompts to extract comprehensive semantic embeddings of labels from large language models; 2) Proposing an Interaction and Fusion Module (IFM) that integrates multiple attention mechanisms to model label co-occurrence and context-aware correlations, fusing label features for more robust representations; 3) Constructing a Quadruplet Mamba-enhanced Visual-Linguistic block (QMVL) to enable bidirectional cross-modal interaction between visual and linguistic features, and adopting input-adaptive category centers based on label representations to boost generalization; 4) Achieving state-of-the-art performance on MS-COCO, Pascal VOC 2007 and NUS-WIDE, with QMVL outperforming attention-based structures in cross-modal interaction efficiency with lower FLOPs and parameters.

Strengths: 1) Fully exploits linguistic modality value by treating it as an equal to visual modality instead of a supplementary; 2) The introduction of Mamba balances cross-modal interaction performance and computational efficiency; 3) Comprehensive ablation studies verify the effectiveness of each module and core design choices; 4) The framework shows good generalization across different backbones and datasets.

Weaknesses: 1) Only validated on general multi-label datasets, with no tests on domain-specific scenarios (e.g., medical imaging); 2) Relies on pre-trained vision-language models, without exploring the impact of pre-training data bias and noise on model performance.

**Audience:**

Yes

**Audience Explanation:**

The paper’s core claims are well-supported by comprehensive, quantitative, and qualitative evidence across rigorous experiments, ablation studies, and comparative analyses.

**Claims And Evidence:**

Yes

**Claims Explanation:**

1. SOTA performance is validated via head-to-head comparisons on MS-COCO, Pascal VOC 2007 and NUS-WIDE across different backbones/resolutions.

2. Module effectiveness is proven by rigorous ablation studies isolating PDLR/IFM/QMVL and their key components.

3. QMVL’s efficiency and superiority are verified by comparisons with attention-based structures (lower FLOPs/params, better performance).

4. Input-adaptive category centers are validated by contrasting with fixed classifier learning on both CLIP and BERT encoders.

5. Robustness is shown via results on noisy NUS-WIDE and qualitative analyses of reduced false predictions vs. baselines.

**Requested Changes:**

1.Add validation on domain-specific multi-label datasets.

2.Empirically analyze pre-trained VLM bias/noise impacts.

3.Fix notation and typographical errors in formulations/figures.

4.Enrich qualitative analysis with diverse scene samples.

---

> ### Author Response · Authors · 2026-03-21
> **Rebuttal for Reviewer QwaN**
>
> ## Q1.Only validated on general multi-label datasets, with no tests on domain-specific scenarios (e.g., medical imaging);
> Thanks for the comments. We add experiments on more domains, including remote sensing and medical imaging, which further verify the efficacy of our method. MVLR consistenly outperforms existing methods in both remote sensing and medical imaging domains, underscoing its robustness in handling domain-specific multi-label classification. The analysis also have been added in the Section A Appendix in the revised manuscript.
>
> Remote sensing (MultiScene[1] and MLRSNet[2]):
>
> | Method        | Backbone | MultiScene |      |      |       |      | MLRSNet |      |      |       |      |
> | ------------- | -------- | ---------- | ---- | ---- | ----- | ---- | ------- | ---- | ---- | ----- | ---- |
> |               |          | mAP        | ALL  |      | Top-3 |      | mAP     | ALL  |      | Top-3 |      |
> |               |          |            | CF1  | OF1  | CF1   | OF1  |         |      |      |       |      |
> | RemoteCLIP-FT | ResNet50 | 66.9       | 63.2 | 73.9 | 52.0  | 65.2 | 97.7    | 90.6 | 92.6 | 67.9  | 70.1 |
> | **MVLR**      | ResNet50 | 67.9       | 64.8 | 75.4 | 52.3  | 66.3 | 98.0    | 91.2 | 92.8 | 66.9  | 70.6 |
> | RemoteCLIP-FT | ViT-B/32 | 67.2       | 63.3 | 73.7 | 52.5  | 65.6 | 97.8    | 91.1 | 93.1 | 66.7  | 70.2 |
> | **MVLR**      | ViT-B/32 | 68.1       | 65.3 | 75.2 | 53.5  | 66.3 | 98.4    | 92.4 | 93.7 | 67.2  | 70.2 |
>
> Medical imaging (ChestX-ray14[3]):
>
> We follow the official data split: 70% of the images are used for training, 10% for validation, and 20% for testing, enabling a fair comparison with existing  methods.
>
> | Method       | Ate      | Car      | Eff      | Inf      | Mass     | Nod      | Pna      | Pnx      | Con      | Ede      | Emp      | Fib      | Pt       | Her      | Mean     |
> | ------------ | -------- | -------- | -------- | -------- | -------- | -------- | -------- | -------- | -------- | -------- | -------- | -------- | -------- | -------- | -------- |
> | CheXNet      | 78.0     | 82.2     | 82.7     | 68.9     | 83.1     | 78.1     | 73.5     | 85.1     | 75.4     | 85.0     | 92.5     | 82.2     | 79.3     | 93.2     | 81.8     |
> | LAAGNet      | 78.3     | 88.5     | 83.4     | 70.3     | 84.1     | 79.0     | 72.9     | 87.7     | 75.4     | 85.1     | 93.9     | 83.2     | 79.8     | 91.6     | 82.4     |
> | CheXGCN      | 78.6     | 89.3     | 83.2     | 69.9     | 84.0     | 80.0     | 73.9     | 87.6     | 75.1     | 85.0     | **94.4** | 83.4     | 79.5     | 92.9     | 82.6     |
> | ADNet        | 80.1     | 89.1     | 84.3     | **71.7** | 84.9     | **82.2** | 75.0     | 89.7     | 76.9     | 86.5     | 94.2     | 95.3     | 80.8     | 96.2     | 83.8     |
> | Anatomy-XNet | 83.1     | **91.4** | 88.6     | **71.7** | 86.0     | 80.4     | 77.1     | 88.2     | 80.9     | 89.9     | 92.9     | 84.4     | 79.8     | **96.4** | 85.1     |
> | GCF-Net      | 83.6     | 89.6     | 88.5     | 70.8     | 88.4     | 80.7     | 79.5     | 90.6     | 82.5     | 91.1     | 92.8     | 83.2     | 82.8     | 95.0     | 85.7     |
> | Ours (rn101) | **84.1** | 90.1     | **89.0** | 71.0     | **88.9** | 81.3     | **79.8** | **91.0** | **83.0** | **91.5** | 93.3     | **83.6** | **83.5** | 95.3     | **86.1** |
>
> [1] Multiscene: A large-scale dataset and benchmark for multiscene recognition in single aerial images.
>
> [2] Mlrsnet: A multi-label high spatial resolution remote sensing dataset for semantic scene understanding.
>
> [3] ChestX-Ray8: Hospital-Scale Chest X-Ray Database and Benchmarks on Weakly-Supervised Classification and Localization of Common Thorax Diseases.

---

> ### Author Response · Authors · 2026-03-21
> **Rebuttal for Reviewer QwaN**
>
> ## Q2. Relies on pre-trained vision-language models, without exploring the impact of pre-training data bias and noise on model performance.
> Thanks for the comments. We tested different pre-trained models, including **MetaCLIP2** (trained with Meta's data), **SigLIP2** (trained with Google's data) and **OpenCLIP** (trained with public data) to explore the potential influence of pre-training data and noise. All compared models are based on ViT-B/16. While noisy pre-training data slightly degrades OpenCLIP's performance, it still achieves competitive results, demonstrating the robustness of the proposed modules. Moreover, the improvements from SigLIP2 primarily stem from its pairwise sigmoid loss instead of a traditional softmax loss, which is inherently more suited for multi-label task. The analysis also have been added in the Section A Appendix in the revised manuscript.
>
> | Method    | MS-COCO |      |      |       |      | NUS-WIDE |      |      |       |      |
> | --------- | ------- | ---- | ---- | ----- | ---- | -------- | ---- | ---- | ----- | ---- |
> |           | mAP     | ALL  |      | Top-3 |      | mAP      | ALL  |      | Top-3 |      |
> |           |         | CF1  | OF1  | CF1   | OF1  |          | CF1  | OF1  | CF1   | OF1  |
> | OpenCLIP  | 89.8    | 84.4 | 85.5 | 78.9  | 80.2 | 68.5     | 65.5 | 75.6 | 60.6  | 70.5 |
> | MetaCLIP2 | 90.7    | 85.0 | 86.7 | 79.8  | 80.8 | 69.0     | 66.4 | 76.1 | 61.9  | 71.4 |
> | SigLIP2   | 91.1    | 85.6 | 87.0 | 80.9  | 82.0 | 69.5     | 67.3 | 77.1 | 62.4  | 72.1 |
> | CLIP      | 90.4    | 84.8 | 86.2 | 79.7  | 81.2 | 68.9     | 66.1 | 76.1 | 61.7  | 71.7 |
>
> ## Q3. Others (qualitative analysis with more scenes, typo fixing, etc.)
> Thanks for the comments. We have added more qualitative analysis on medical images and remote sensing samples (please see Figure 7 in our revised manuscript). We also proofread our paper and fix any notation and typographical errors in the formulations and figures.

---

> > ### Comment · Reviewer_QwaN · 2026-03-26
> > **Thank you to the authors for your response**
> >
> > I have reviewed all the revision requests I previously raised, and confirm that all issues have been properly addressed.

---

### Decision · Action_Editor_gBzj · 2026-06-04

**Recommendation:** Accept with minor revision

**Additional Comments:**

This paper has undergone a rigorous review process with three reviewers raising substantive concerns, all of which have been adequately addressed through revision and rebuttal. The core contributions — PDLR, IFM, and QMVL — are clearly motivated and empirically validated. The authors have substantially expanded the manuscript with additional experiments, efficiency profiling, domain generalization tests, and comparison with Mamba-based methods including MambaML (ICCV 2025).
Prior to final acceptance, the authors should revise the paper ensuring all the discussions and additional evidence provided during rebuttal are incorporated in the final version. Especially, please ensure the following changes -  (1) the motivation for the H1–H4 quadruplet construction in QMVL, as explained in the rebuttal, is reflected in the main text; (2) all typographical errors identified during review are corrected; (3) the rationale for using the CLIP visual encoder — and the comparisons with different-pretrained methods — is addressed in the main experimental section.

**Audience:**

Yes

**Audience Explanation:**

This work is at the intersection of several active research areas — vision-language models, prompt tuning, state space models, and multi-label recognition — all of which are of broad interest to the TMLR community. The paper's core contribution of treating linguistic representations as an equal modality (rather than a supplementary signal) and fusing them via a Mamba-based cross-modal block is a meaningful architectural contribution. The consistent performance gains across multiple backbones, resolutions, and datasets, as well as the extended validation in remote sensing and medical imaging, make the findings useful to researchers and practitioners working on multimodal recognition tasks.

**Claims And Evidence:**

Yes

**Claims Explanation:**

The claims made in this submission are well-supported by the time of the final revised manuscript. The paper demonstrates state-of-the-art performance on three standard multi-label recognition benchmarks (MS-COCO, Pascal VOC 2007, NUS-WIDE) through comprehensive quantitative comparisons. Ablation studies isolate the contribution of each proposed component (PDLR, IFM, QMVL), confirming that improvements are not attributable to backbone choice alone. Following a thorough review and rebuttal process, the authors have additionally provided: (1) efficiency comparisons including FLOPs, parameters, inference latency, and GPU memory; (2) comparisons with recently published Mamba-based baselines (MambaML, MambaVision, SiMBA) and CLIP-based methods (PatchCT); (3) experiments on domain-specific datasets (remote sensing and medical imaging); and (4) ablation of the QMVL design against attention-based alternatives. Reviewers QwaN and kinT confirmed their concerns were addressed. While Reviewer WM6Y raised substantive concerns about theoretical motivation for the quadruplet Mamba design, the authors provided a principled explanation of the H1–H4 construction in terms of bidirectional cross-modal conditioning under linear-complexity constraints. The overall evidentiary basis is sufficient for acceptance.

---

> ### Author Response · Authors · 2026-07-02
> **Response to the Decision Letter**
>
> Dear Editor,
>
> We sincerely thank you for the positive decision and for the constructive final comments. We have carefully revised the manuscript according to the AE’s suggestions and incorporated the relevant discussions and evidence from the rebuttal into the final version.
>
> Specifically, we made the following changes:
>
> 1. We clarified the motivation for the H1-H4 quadruplet construction in QMVL in the main text, explaining how it supports bidirectional visual-linguistic interaction under the Mamba framework.
>
> 2. We carefully checked the manuscript and corrected the typographical errors identified during review.
>
> 3. We updated Section 4.2 to explain the rationale for using CLIP as the default visual encoder, since MVLR requires aligned visual and linguistic features. We also explicitly refer to Appendix Table 15, where comparisons with different pre-trained vision-language models are reported.
>
> We hope the revised manuscript fully addresses the remaining concerns. Thank you again for your time and guidance throughout the review process.
>
> Sincerely,
> The Authors